# Bow shock oscillations of Mars under weakly disturbed solar wind conditions

Long Cheng [1,2,3], Yuming Wang [1,3,4] ✉, Yingjuan Ma[5], Robert Lillis [2], Jasper Halekas[6], Benoit Langlais [7], Tielong Zhang[3,8,9], Aibing Zhang[10,11], Guoqiang Wang[9], Sudong Xiao[9], Zhuxuan Zou[1,3], Yutian Chi[12], Xinjun Hao[1,3], Yiren Li [1,3], Zonghao Pan[1,3] & Kai Liu[1,3]

Bow shock, where the solar wind first encounters the Martian environment, reflects the complex interplay between the solar wind and Martian upper atmosphere and crustal fields. However, a comprehensive understanding of Martian bow shock dynamics remains elusive due to limited multi-spacecraft observations. Here, leveraging the joint observations from China's Tianwen-1 and NASA's Mars Atmosphere and Volatile EvolutioN (MAVEN), we reveal Martian bow shock oscillations with a temporal scale of minutes and spatial extents of hundreds of kilometers during weakly disturbed solar wind. Our analysis of the observations along with three-dimensional simulations suggests that magnetosonic Mach number is the most sensitive parameter influencing the bow shock, and a slow solar wind stream that favors low Mach numbers may lead to the large-scale bow shock oscillations and the whole Martian space environment. This finding advances our understanding of the interactions between the solar wind and non-magnetized planets.

Exploring and understanding Mars' complex space environment and atmospheric escape mechanisms are focuses across past, ongoing and future Mars missions[1–6]. Moving from the Sun towards Mars in its near-space environment, one encounters various plasma boundaries and regions[7–12]. First is the bow shock (BS), behind which is the turbulent magnetosheath, followed by the so-called induced magnetosphere boundary or the magnetic pileup boundary (MPB). Behind the MPB lies the pilot region and then the ionosphere, which are separated by the photoelectron boundary. As the first interface at which solar wind plasma enters the Martian magnetosphere, the BS and its behavior are important to investigate[13]. Its shape and position reflect the balance between the external heliospheric environment governed by the solar wind and interplanetary magnetic field (IMF) and the magnetosphere influenced by the Martian ionosphere and the crustal magnetic fields. A change of the BS shape and/or position may reflect either a change in external conditions or internal processes, e.g., the possible magnetic reconnection of the crustal field with the draped IMF[14–16], the large disturbance or enhanced ion escape during global dust storms[17].

Long-term observations from the Mars Global Surveyor[18], Mars Express[19], and MAVEN[20] missions have revealed that the global shape and location of the BS are controlled by the solar extreme ultraviolet (EUV) irradiance, the solar wind dynamic pressure, the magnetosonic

[1]National Key Laboratory of Deep Space Exploration/School of Earth and Space Sciences, University of Science and Technology of China, Hefei, China. [2]Space Sciences Laboratory, University of California, Berkeley, CA, USA. [3]CAS Center for Excellence in Comparative Planetology/CAS Key Laboratory of Geospace Environment/Mengcheng National Geophysical Observatory, University of Science and Technology of China, Hefei, China. [4]Hefei National Laboratory, University of Science and Technology of China, Hefei, China. [5]Department of Earth, Planetary, and Space Sciences, University of California, Los Angeles, CA, USA. [6]Department of Physics and Astronomy, University of Iowa, Iowa City, IA, USA. [7]Nantes Université, Univ Angers, Le Mans Université, CNRS, Laboratoire de Planétologie et Géosciences, Nantes, France. [8]Space Research Institute, Austrian Academy of Sciences, Graz, Austria. [9]Institute of Space Science and Applied Technology, Harbin Institute of Technology, Shenzhen, China. [10]National Space Science Center, Chinese Academy of Sciences, Beijing, China. [11]University of Chinese Academy of Sciences, Beijing, China. [12]Institute of Deep Space Sciences, Deep Space Exploration Laboratory, Hefei, China. ✉e-mail: ymwang@ustc.edu.cn

Mach number, the IMF, and the crustal fields[21,22]. Various statistical models for the BS were also established based on data from different solar activity periods[23]. However, dynamics of the Martian BS on much shorter time-scales and their causes are not yet well understood, requiring simultaneous monitoring of the upstream solar wind and the IMF[24]. The instant responses of the BS to disturbances could be its motion, reformation, rippling or occurrences of some other transient structures[25–30]. Although the dynamics of Earth's BS, and its relationship with upstream solar wind conditions have been extensively studied with multi-spacecraft missions, significant differences in its behavior could be expected for Mars due to the lack of an intrinsic global magnetic field.

China's first Mars mission Tianwen-1[31] carries a magnetometer and an ion and neutral particle analyzer on its orbiter, called MOMAG[32,33] and MINPA[34], collecting magnetic field and ion data since 13 November 2021. Tianwen-1 orbiter has a large elliptical orbit (265 × 12000 km altitude) which frequently crossed the flank of the BS on the dawn/dusk side during the last 1.5 months of 2021 (see Supplementary Notes and Supplementary Fig. 1). While Tianwen-1 is near the BS, MAVEN is sometimes upstream of the shock, monitoring the solar wind. The resulting unprecedented combination of high-resolution measurements from Tianwen-1 and MAVEN provides unique insights into the characteristics and dynamics of the Martian BS.

Here, we show the observational evidence of the global oscillations of Martian BS during weakly disturbed solar wind conditions by two events. The multiple BS crossings of Tianwen-1 reveal that these BS oscillations are at the time-scale of minutes and spatial-scale of hundreds of kilometers. By comparing with single BS crossing events, we find that such BS oscillations tend to occur at low magnetosonic Mach number, suggesting that Mach number is a good parameter assessing the susceptibility of the Martian BS, and a weaker BS is more easily oscillated by solar wind disturbances than a stronger BS. These results are further confirmed by our three-dimensional numerical simulations.

## Results

### Observations of the minute-scale large oscillations

During the period from 13 November to 31 December in 2021, Tianwen-1 recorded single and multiple BS crossing events[35,36] during its inbound or outbound one-way trajectory (see Supplementary Fig. 2). Here, to avoid ambiguity, we define that for a complete crossing of BS, the spacecraft should stay in upstream and downstream for at least one minute. Moreover, we focus on the multi-crossing events under quasi-steady or weakly disturbed solar wind conditions that is particularly interesting and will be illustrated later. After inspecting the data and removing all the other possibilities[37,38], e.g., hot flow anomalies, sheath jets, etc., we report two confirmed BS oscillation events below.

The first event (event 1) occurred between 05:25 and 05:50 UT on 2 December 2021 (Fig. 1), when Tianwen-1 was crossing the BS from the solar wind to the magnetosheath, with |B| increasing from ~5 to ~13 nT. During the period of this crossing, we find three large jumps in magnetic field around 05:36 UT ($t_2$), 05:40 UT ($t_3$) and 05:42 UT ($t_4$), respectively, as denoted by the vertical dashed lines in the figure. At each jump, the value of |B| changed between the levels of the IMF and the sheath. Between $t_2$ and $t_3$, |B| stayed around 15 nT for about 4 min, while between $t_3$ and $t_4$, |B| stayed around 5 nT for about 2 min. There was also a jump in the magnetic field around 05:33 UT ($t_1$), after which the magnetic field climbed up to ~13 nT in a minute and then gradually return to the IMF level in another minute. This variation may indicate a brief encounter with the BS but not a complete crossing of the BS. Our following analysis focuses on the jumps at $t_2$, $t_3$, and $t_4$. We believe that these three jumps reflect the large-scale, rapid motion (or oscillation) of the BS across the relatively slow-moving spacecraft, three distinct times. The observational features of this event could be distinguished from other featured structures near the shock surface, as analyzed below.

Figure 2 summarizes the featured structures that could be observed near a planetary BS[37,38], including: (1) various transient structures in the foreshock region ahead of the Martian BS, e.g., hot flow anomalies, spontaneous hot flow anomalies, foreshock bubbles, foreshock cavities, foreshock cavitons, density holes, foreshock compressional boundary, and short large-amplitude magnetic structures (see Supplementary Notes for details), (2) sheath jets and magnetic holes in the downstream of the shock, and (3) ripples on the shock surface[28]. Those events may not only lead to variations in the magnetic field but also cause changes in the ion energy spectrum. For magnetic field changes on the order of minutes and ion flux changes at similar time scales and energies near 1 keV, MOMAG and MINPA are capable of detecting these changes (see Supplementary Notes for details).

For the event shown in Fig. 1, not only the magnetic field but also the measured ion fluxes (Fig. 1c) change between the levels of the solar wind and sheath. When Tianwen-1 was in the solar wind before $t_2$, |B| was around 5 nT and the ion data from MINPA showed nearly no solar wind signal, due to its limited field of view (FOV) (see "Methods", subsection "Distinguishing solar wind and magnetosheath from MINPA data", and Supplementary Fig. 4 for details). Between $t_2$ and $t_3$, along with the significant increasing of |B|, the detected ion fluxes increased and the distribution broadened due to the heat of plasma and deflection and scattering of solar wind protons into MINPA's FOV, which is characteristics of magnetosheath. Between $t_3$ and $t_4$, |B| returned to the IMF level and MINPA detected no signal again, suggesting that Tianwen-1 returned to the solar wind. This behavior is quite different from those of the transients demonstrated in Fig. 2, of which the magnetic field and ion fluxes show no changes in such a synchronous way between the levels of the solar wind and magnetosheath at timescales of minutes (see Table 1).

However, the rippling of the BS[28] cannot be uniquely ruled out by the above analysis. When a spacecraft crosses a rippled BS, it may cross the shock front several times, which can also cause |B| and ion fluxes to vary between the levels of the solar wind and magnetosheath, i.e., the same signature caused by the large-scale motion of the shock. It is noticed that the shock normal direction varies along the rippled shock surface, but not along an oscillated large-scale shock surface[39]. We estimated the shock normal directions for individual crossings using minimum variance analysis (MVA), and found small changes (<10°) in the normal (see Supplementary Tables 1 and 2). We also assessed the overall shock normal direction (see "Methods", subsection "BS normals") for the entire event to determine if the strength of the magnetic field component along the shock normal varied[28]. Our analysis revealed that the normal magnetic field was indeed quite stable (see the magenta line in Fig. 1a). Notably, there were no significant changes in the downstream region around each crossing. Therefore, we exclude the possibility that the multiple crossings were caused by shock ripples.

Additionally, the transient structures demonstrated in Fig. 2 are typically associated with quasi-parallel shocks, where the angle between the IMF and shock normal, $\theta_{Bn}$, is less than 45°. In the two events studied in this work, $\theta_{Bn}$ were as large as 70° and 87°, respectively (see Supplementary Table 3). Hence, both events are classified as quasi-perpendicular shocks, further excluding the likelihood of transient events that primarily occur near quasi-parallel shocks.

Besides, the possibility of Kelvin–Helmholtz (KH) instability is also considered. However, it is noticed that the KH instability is found to frequently occur at the magnetopause and ionopause[40,41], but there is no evidence that it may cause BS oscillations. If KH instability were a driving factor, frequent multi-crossings of the BS would be expected, which is not the case in observations. Even if KH happened on the BS surface, it still can be ruled out due to the minimal changes in the shock normal obtained above.

Based on the above analysis, we may conclude that when Tianwen-1 moved from the solar wind to the sheath, the Martian BS oscillated

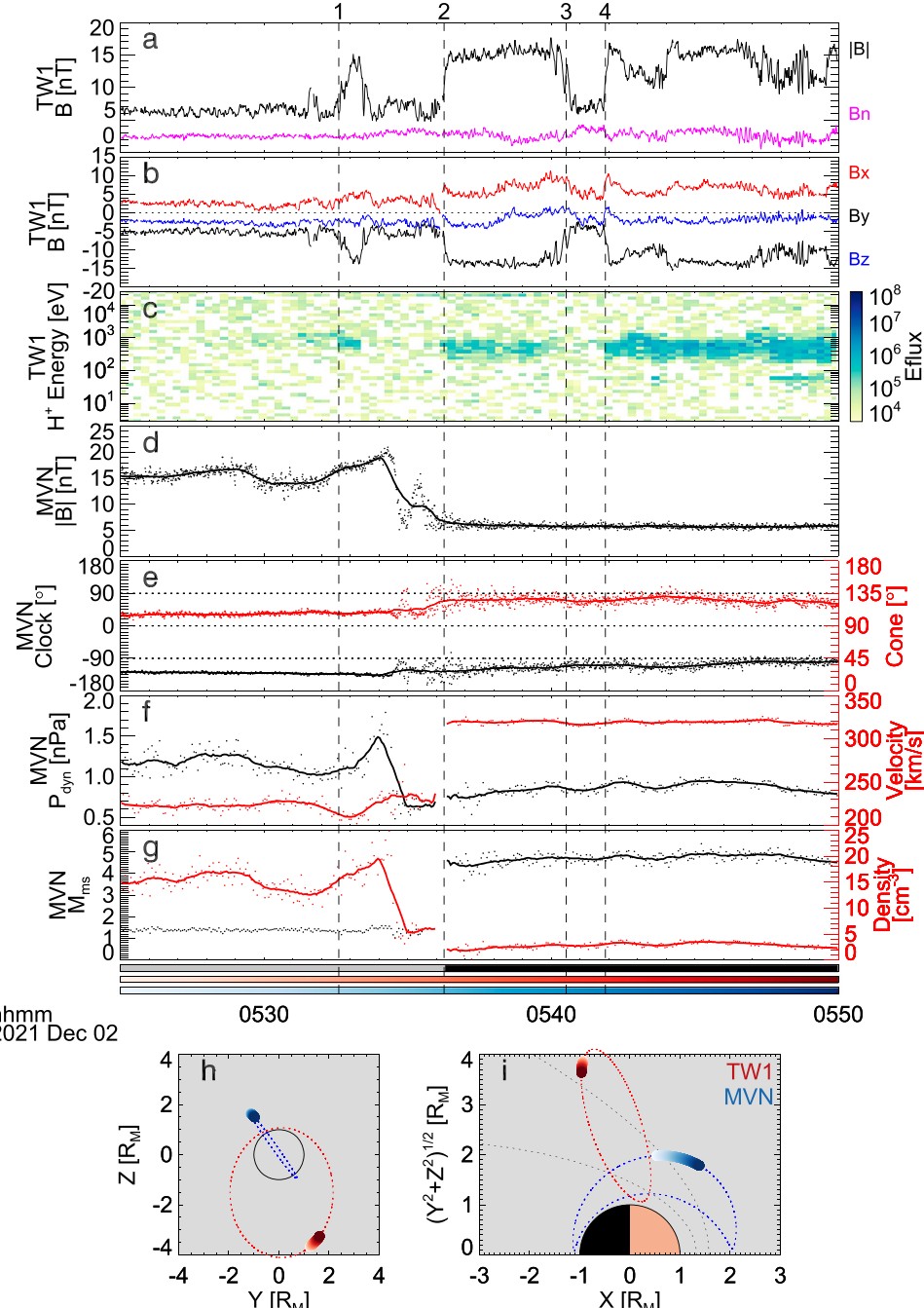

**Fig. 1 | Observations of the bow shock oscillation event 1. a** The magnetic field strength (black) from MOMAG 1 Hz data and the normal magnetic field (magenta). **b** Three components of the magnetic field from MOMAG in Mars-Solar-Orbital (MSO) coordinates. **c** Proton energy spectra measured by MINPA, with colors representing the energy flux in eV/(eV cm² s sr). **d** The magnetic field strength from MAVEN/MAG 1 Hz data. **e** Directions of the magnetic field from MAVEN/MAG. **f** The solar wind dynamic pressure and velocity. **g** The magnetosonic Mach number and density. **h** Positions of Tianwen-1 and MAVEN in the MSO Z-Y plane. **i** positions of Tianwen-1 and MAVEN in the MSO cylindrical coordinates, where locations of the bow shock and magnetic pileup boundary by Edberg et al.[59] are shown for reference. The mode of SWIA operation is shown by the black-gray color bar, with black for the solar wind mode and gray for the sheath mode. The switch between the two modes may cause unphysical discontinuities in parameters derived from SWIA data. The red color bar and blue color bar represent the time for the spacecraft location shown in (**h**) and (**i**), with red for Tianwen-1 and blue for MAVEN. The dots in **d**–**g** show the original data from MAVEN, while the solid lines illustrate the 60-s smoothed results.

quickly, causing Tianwen-1 to cross the BS three times at $t_2$, $t_3$ and $t_4$, respectively. Further, we can calculate the spatial scale of the oscillation along the normal direction (see Methods, subsection BS normals), which is about 150 km (between $t_2$ and $t_4$), the lower limit of the real oscillation scale. During the period from $t_2$ to $t_4$, MAVEN stayed in the solar wind. It recorded fluctuations at the minute scale after $t_2$ in the IMF strength, IMF direction, solar wind velocity, density, dynamic pressure and Mach number as indicated by the solid lines in Fig. 1d–g. These fluctuations were persistent during the whole event, and there was no extra-large fluctuation right before each BS crossing. Thus, we call it quasi-steady or weakly disturbed solar wind. It should be noticed that although some solar wind parameters fluctuated significantly at second scale, they cannot be the reason for the minute-scale oscillations. These observations imply that either other factors/processes

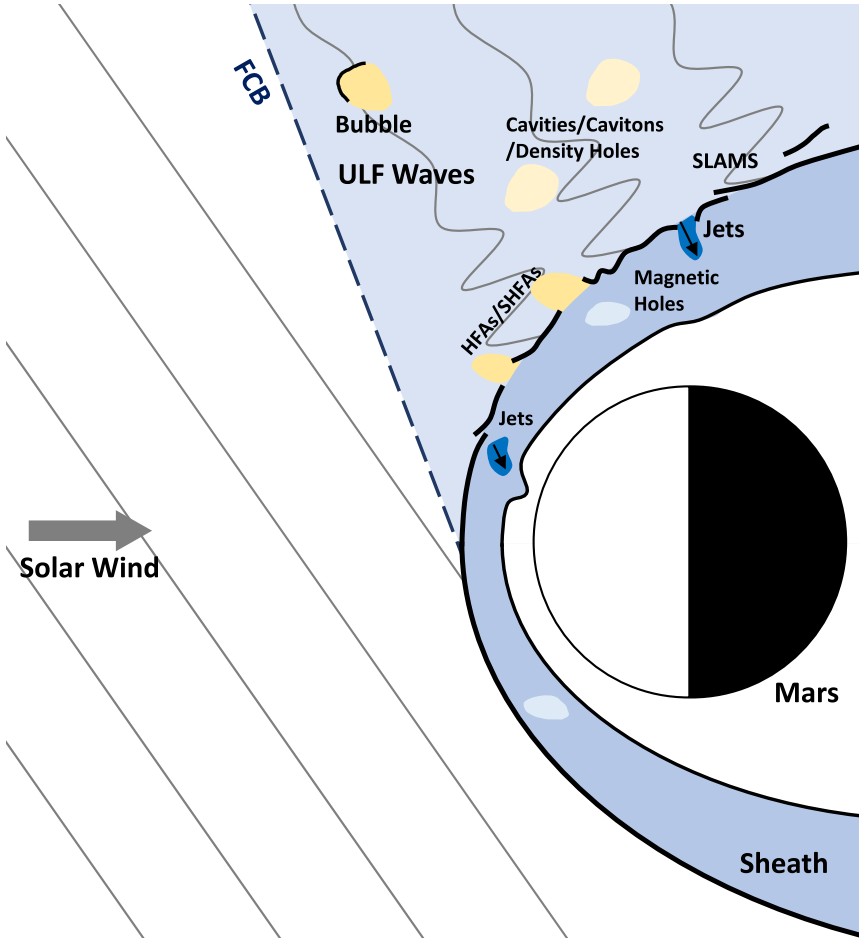

**Fig. 2 | Schematic illustrations of the transients near the bow shock.** The cartoon shows the transients upstream and downstream of the bow shock, including the hot flow anomalies (HFA), spontaneous hot flow anomalies (SHFA), foreshock bubbles, foreshock cavities, foreshock cavitons, density holes, foreshock compressional boundary (FCB), short large-amplitude magnetic structures (SLAMS), sheath jets and magnetic holes.

## Table 1 | Transients near the bow shock

| Region | Transients | Duration (Earth)[35] | Duration (Mars) | Reference figure |
|---|---|---|---|---|
| Foreshock | (Spontaneous) Hot flow anomalies | Minutes | ~1 min[66,67] | Collinson et al.[66] |
| | Foreshock bubbles | Minutes | ~1 min[68] | Madanian et al.[68] |
| | Foreshock cavities/ Foreshock cavitons | Minutes | Minutes[69] | Collison et al.[69] |
| | Foreshock compressional boundary | Minutes | Seconds[69] | Collison et al.[69] |
| | Density holes | Seconds | | Parks et al.[70] |
| | Short large-amplitude magnetic structures (SLAMS) | ~10 s | | Plaschke et al.[71] |
| Sheath | Sheath jets | Minutes | Minutes[72] | Plaschke et al.[73] |
| | Magnetic holes | Milliseconds to Minutes | | Plaschke et al.[73] |

Durations of transient structures at Earth are compiled from Zhang et al.[37], while durations at Mars are referenced from previous studies at Mars[66–69,72]. References to representative figures (not shown here) for each type of transient are also provided.

beyond external solar wind variability may cause the BS to oscillate, or the Martian BS is in a special state when even weak disturbances of solar wind can drive such large-scale oscillations.

The second event (event 2) is shown in Fig. 3. Tianwen-1 observed the oscillation of the BS during 20:15−20:40 UT on 25 December 2021.

As revealed by the variations of the magnetic field and ion fluxes during this period, Tianwen-1 entered the sheath from the solar wind. Tianwen-1 was in the solar wind before ~20:20 UT ($t_1$), as |**B**| measured by MOMAG was steady at ~3 nT and MINPA missed the solar wind ion beam. Between $t_1$ and 20:25 UT ($t_2$), Tianwen-1 was in the foreshock, as

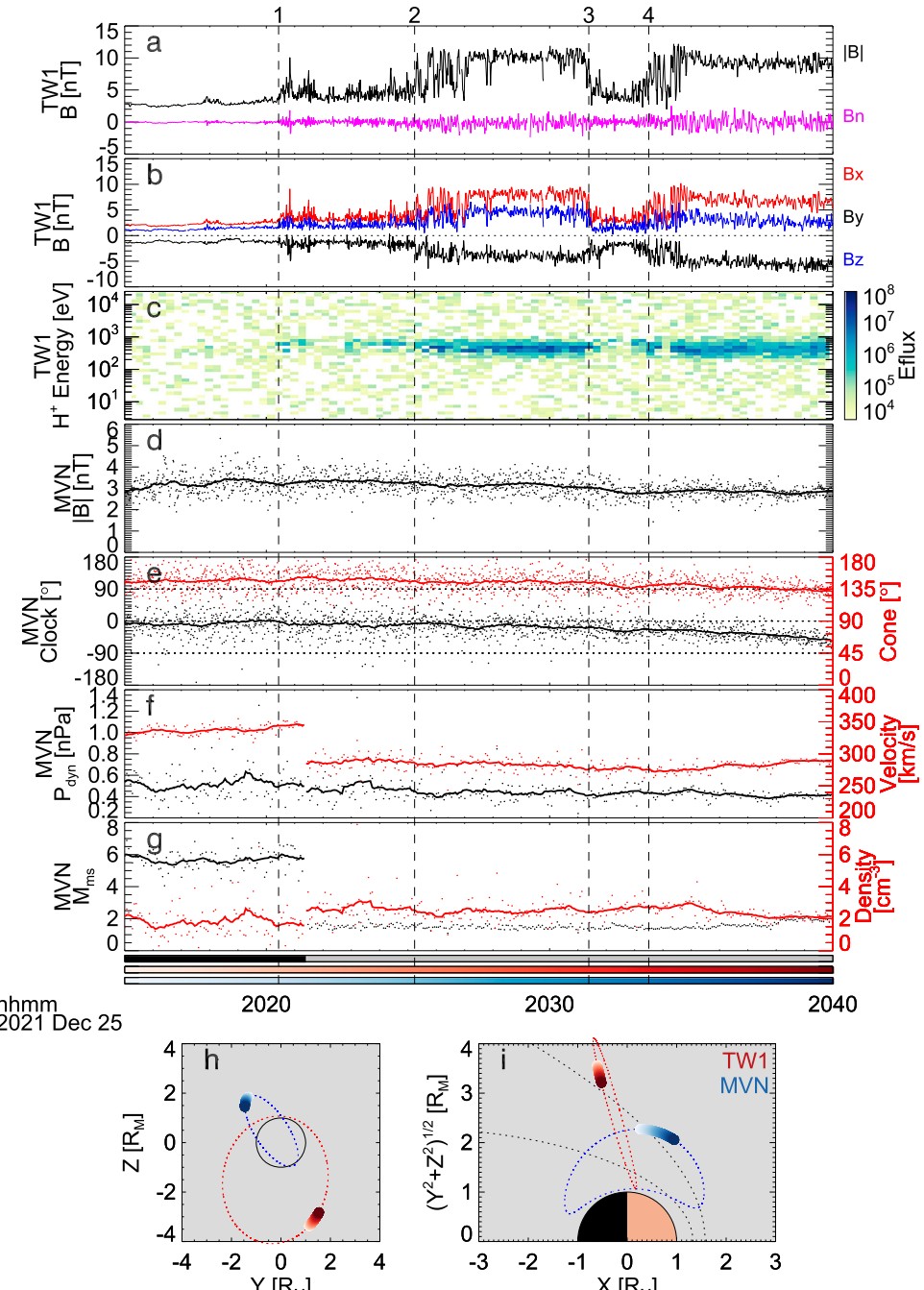

**Fig. 3 | Observations of the bow shock oscillation event 2. a** The magnetic field strength (black) from MOMAG 1 Hz data and the normal magnetic field (magenta). **b** Three components of the magnetic field from MOMAG in MSO coordinates. **c** Proton energy spectra measured by MINPA, with colors represent the energy flux in eV/(eV cm² s sr). **d** The magnetic field strength from MAVEN/MAG 1 Hz data. **e** Directions of the magnetic field from MAVEN/MAG. **f** The solar wind dynamic pressure and velocity. **g** The magnetosonic Mach number and density. **h** Positions of Tianwen-1 and MAVEN in the MSO *Z-Y* plane. **i** Positions of Tianwen-1 and MAVEN in the MSO cylindrical coordinates. The layout and symbols are the same as those in Fig. 1.

the magnetic field fluctuated and the ion fluxes were slightly enhanced. The first BS crossing occurred at $t_2$, with |**B**| increasing to ~10 nT and also an enhancement of ion fluxes around 500 eV. Between $t_3$ and $t_4$, the magnetic field returned to the IMF level and the ion flux was weak, which indicated Tianwen-1's return to the solar wind. At $t_4$, Tianwen-1 entered the sheath again, with |**B**| and ion fluxes increasing. During this period, the magnetic field strength along the shock normal was relatively stable, especially in the downstream around each crossing (see the magenta line in Fig. 3a). The synchronous variation of the magnetic field and the ion flux between the levels of the IMF and sheath and steady normal magnetic field indicates the BS oscillation in this event.

The oscillation amplitude along the normal direction is about 400 km (between $t_2$ and $t_4$).

During the full period of interest, MAVEN was in the solar wind, as shown by the orbit map (Fig. 3i) and plasma measurements (Fig. 3d–g). The IMF strength, clock angle and cone angle fluctuated around 3 nT, 0°, and 135°, respectively. The solar wind dynamic pressure, $P_{dyn}$, fluctuated around 0.4 nPa, with some second scale spikes. The magnetosonic Mach number was also stable, except for the discontinuity cause by the mode switch of the Solar Wind Ion Analyzer (SWIA)[42] from the solar wind mode to the sheath mode at ~20:21:30 UT (see the black-gray color bar in Fig. 3). During the period of this event, the solar wind

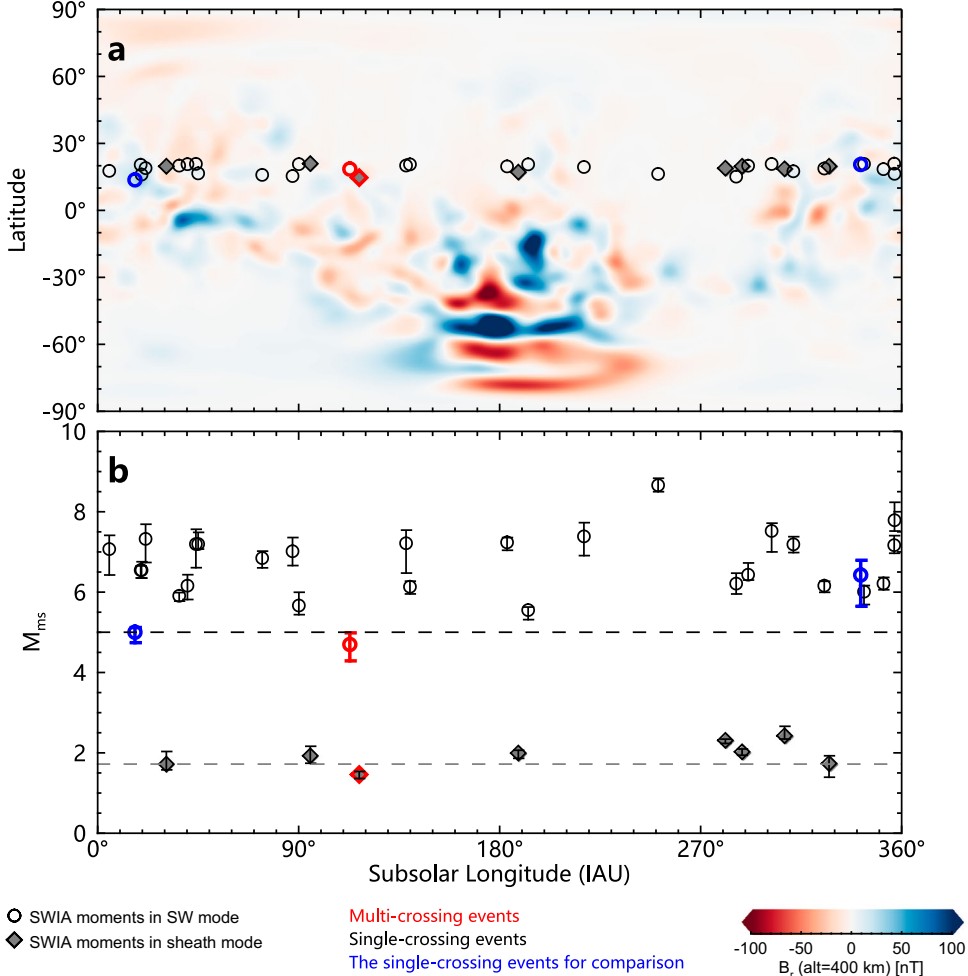

**Fig. 4 | Distributions of subsolar points of the BS nose and magnetosonic Mach number of the solar wind during the events of interest. a** Locations of the subsolar points, with the background showing the radial component of the crustal magnetic field at 400 km altitude, derived from the crustal field model by Langlais et al.[65]. **b** Distributions of the magnetosonic Mach numbers of the solar wind with error bars versus the longitudes of the subsolar points in the IAU-Mars coordinate system. The circle and diamond symbols represent the SWIA mode with the circle indicating solar wind mode and the diamond indicating sheath mode. Red, black, and blue symbols correspond to multi-crossing events under weakly disturbed solar wind conditions, the single-crossing events, and the single-crossing event selected for comparison with a numerical simulation, respectively. Specifically, the red circle represents event 1 and the red diamond event 2. Source data are provided as a Source Data file.

parameters similarly showed persistent minute-scale disturbances but no particularly extra-large one at the BS crossings, suggesting that the solar wind was in quasi-steady or weakly disturbed state during the BS crossings of Tianwen-1.

## Cause of the oscillations

As previously suggested, there are two possible reasons for such BS oscillations. One is some internal processes[22] and the other is still attributed to external solar wind conditions[21,35]. We first check the possibility of the internal cause. Magnetospheric oscillations induced by internal drivers have been documented across magnetized celestial bodies, such as Earth and Saturn. Saturn, in particular, commonly exhibits oscillations in its BS and magnetopause aligned with the planetary period, potentially arising from co-rotating sources near the planetary region[43,44]. The propagation of plasma waves generated by atmospheric phenomena is also considered a contributing factor to the extensively studied Saturnian planetary period oscillations within its magnetosphere[45–47], potentially influencing circulation patterns and mass loss within Saturn's magnetosphere[48]. On Earth, ionospheric outflow may induce magnetospheric oscillations[49].

For non-magnetized planets, like Mars, previous studies demonstrated that the Martian crustal field can affect the solar wind flow and the interaction between the solar wind and Martian space environment[50,51]. The Martian crustal field may increase the local altitude of the BS, with a dependence on the angular extent[22], i.e., the angle between the BS location and the crustal magnetic field with respect to the Mars center. Therefore, it is reasonable to speculate that the strong crustal magnetic fields rotating with Mars could also influence the dynamics of the BS, leading to its oscillation during quasi-steady solar wind conditions. Therefore, we investigate the longitudes of the subsolar points during the events. It is found that the subsolar longitudes during the two events were around 115° in the Martian geographical coordinates (see the red symbols in Fig. 4a), which is near the west boundary of the Martian strongest magnetic anomaly region.

For comparison, we examined 35 single-crossing events observed by Tianwen-1 during the same period from 13 November to 31 December in 2021 (see Supplementary Table 4). These events were selected based on the criteria of (1) the clear signature of the BS crossing in the magnetic field from Tianwen-1, (2) no notable fluctuations blurring the distinction between single- and multiple-crossings, and (3) availability of the simultaneous observations of the upstream solar wind from MAVEN. We find that the BS subsolar longitudes during these events are nearly uniformly distributed (the black symbols in Fig. 4a), which means there were also single-crossing events

when the subsolar longitudes were around the west boundary of the Martian strongest magnetic anomaly region. It suggests that the Martian crustal field is not likely the cause, or at least not the only cause, for the observed BS oscillations.

A possible scenario involving the Martian crustal field is that the dusk-side located southern crustal field may favor magnetic reconnection under some IMF conditions and therefore cause notable disturbances on the BS. Recent studies[52–55] on discrete auroras at Mars have suggested that enhanced magnetic reconnections prefer to occur when the strongest crustal fields are located on the duskside and the IMF has a -$B_y$ orientation (i.e., the IMF has a negative clock angle). At that configuration, the large shear angle between the draped IMF and the local crustal magnetic field is easier to trigger the reconnection of magnetic fields. Then we examine the IMF conditions, and particularly notice the events of which the subsolar longitude is within 70–150° (or the strongest crustal field region is located on the duskside). It can be observed that for the two oscillation events, the IMF had a -$B_y$ orientation. However, there were also three single-crossing events with a duskside-located strongest crustal field region and -$B_y$ IMF orientation (see Supplementary Fig. 3a). This result suggests again that the Martian crustal field is not likely the cause of the observed BS oscillations.

We then return back to the external solar wind conditions, and especially examine the solar wind density, dynamic pressure, magnetosonic Mach number and IMF strength. We use the 60-s smoothed data in a 10-min time window centering on a BS crossing to calculate the median value and the difference between the maximum and minimum values. The ratio of the max-min difference to the median value is treated as the disturbance level. By putting the two multiple-crossing events and 35 single-crossing events together, we find that the magnetosonic Mach number rather than the disturbance level plays a key role (see Fig. 4b and Supplementary Fig. 3b–e). The two BS oscillation events have the lowest Mach number in their own sample sets. The solar wind during event 1 and 28 single-crossing events was measured in the solar wind mode by MAVEN/SWIA (indicated by circles in Fig. 4b), which measures the solar wind ion flux with a narrower FOV and a higher angular resolution than the sheath mode. The Mach number of event 1 is about 4.7, lower than those of all the 28 single-crossing events, which are at least 5.0 (indicated by the upper dashed line in Fig. 4b).

For event 2 and the other 7 single-crossing events, solar wind was measured in the sheath mode (the filled diamonds in Fig. 4b). In this mode, the solar wind velocity is underestimated and the density slightly overestimated, which eventually lead to the underestimation of the magnetosonic Mach number. Nevertheless, such inaccuracies are systematic and will not affect the comparison of events in the same mode. It is found that the Mach number of event 2 is about 1.5, also lower than those of the 7 single-crossing events, which are above 1.7 (indicated by the lower dashed line in Fig. 4b). No similar pattern is found in the solar wind density and dynamic pressure (see Supplementary Fig. 3c, d). Mach number is a measure of the shock strength. Low Mach number means a weak BS. Thus, our analysis shows the observational evidence that a weaker BS is more easily oscillated by solar wind disturbances than a stronger BS.

The above conclusions are further confirmed by three-dimensional magnetohydrodynamic numerical simulations[47,49] (see "Methods", subsection "3D numerical simulations"). Two simulation cases on event 1 are carried out as shown in Fig. 5. One uses the observed solar wind condition as input (Case 1), and the other uses the observed solar wind condition but with Mars rotated forward 90° to check the effect of the crustal field (Case 2). The Case 1 simulation clearly shows the features of large-scale BS oscillations (Fig. 5b, c), though the magnetic field profile along Tianwen-1 trajectory does not exactly match the observations due to the limited spatial and temporal resolutions of the simulations (see Methods, subsection Distinguishing solar wind and magnetosheath from MINPA data). Case 2 shows the almost identical result (the orange line in Fig. 5b), confirming the

previous conclusion that the crustal field is not the cause of the observed BS oscillations.

Additionally, we perform simulations on two single-crossing events, i.e., Events 12 and 37 (see Supplementary Figs. 6 and 7, and the blue symbols in Fig. 4 and Supplementary Fig. 3), as comparisons. Event 12 is selected because Tianwen-1 crossed the BS at a location similar to event 1 and the solar wind disturbance level was larger than that of Event 1. Event 37 is selected because the Mach number is 5.0, just a little higher than that of Event 1. Both simulations do not show notable multiple oscillations of the BS (Fig. 5d–g). For event 12, the simulated BS position shown in Fig. 5e shows one weak oscillation of ~0.1 $R_M$ around 04:42, much smaller than the multiple oscillations of about 0.2 $R_M$ shown in Fig. 5c for event 1. After then, the BS of event 12 continuously shrinks, leading to a single crossing event recorded by Tianwen-1. For event 37, the simulation shows even smaller changes in BS position. All the simulations confirm the key role of Mach number.

In summary, the study reports the direct observations of the minute- and hundred-kilometer-scale oscillations of Martian BS under weak solar wind disturbances. The comparison to other single-crossing events shows that the Mach number is a good parameter assessing the susceptibility of the Martian BS. We can therefore imagine that a slow solar wind stream that favors low Mach numbers may lead to the large-scale disturbances of Martian BS and the whole Martian space environment.

## Methods
### Upstream parameters
The solar EUV irradiance and the upstream solar wind condition are observed by MAVEN. The solar EUV irradiance observed by the solar EUV monitor[56] of MAVEN are in 0.1–7 nm, 17–22 nm, and 121–122 nm bands. The EUV irradiances are not shown in Fig. 1 and Fig. 3, due to their stability across tens of minutes in these cases. The clock angle of the IMF in the MSO coordinate system is calculated by $\tan^{-1}(B_y/B_z)$, which results in 0°, 90°, 180° and −90° for the +Z, +Y, −Z and −Y, respectively. The cone angle is calculated by $\cos^{-1}(-B_x/|\mathbf{B}|)$, which is the angle between the magnetic field vector and the Sun-Mars direction and results in 0° and 180° for the −X (Mars-ward) and +X (sunward), respectively. The magnetosonic Mach number is calculated by

$$M_{ms} = v_{sw}/\sqrt{c_s^2 + v_A^2},$$ with the sonic speed $c_s = \sqrt{(T_e + 5/3\,T_p)/m_p}$ and the Alfven speed $v_A = |\mathbf{B}|/\sqrt{\mu_0 \rho}$, where $v_{sw}$ and $\rho$ are the speed and mass density of the solar wind, $T_p$ is the proton temperature from the Solar Wind Ion Analyzer[42] of MAVEN, and the electron temperature $T_e$ is assumed to be the same as $T_p$. The solar wind dynamic pressure is calculated by $P_{dyn} = \rho v_{sw}^2$.

### BS normals
The shock normal can be estimated from the time-series data measured by the spacecraft, based on the coplanarity theorem[57] or MVA[58]. The mix mode normal based on the coplanarity theorem requires the magnetic field and ion velocity upstream and downstream of the shock[57], where the latter is hard to be obtained during the periods of the events in this work, due to the limited FOV of MINPA (see the next subsection for details). Here we determine the BS normal using MVA, $\mathbf{n}_{MVA}$, which identifies the direction along which the magnetic field exhibits minimal variation, assuming that the magnetic field remain the shock normal keeps constant[57]. We apply the MVA method on the 1 Hz magnetic field data during the entire period of the event of interest. In other words, for the event shown in Fig. 1, the data from 05:25 UT to 05:50 UT are used, and for the event shown in Fig. 2, the data from 20:15 UT to 20:40 UT are used.

The BS normal, $\mathbf{n}_{mod}$, can also be estimated by the conic BS model. Typically, the BS is assumed to be symmetric around the 4°-aberrated MSO X axis, respecting to the orbital velocity of Mars and the solar

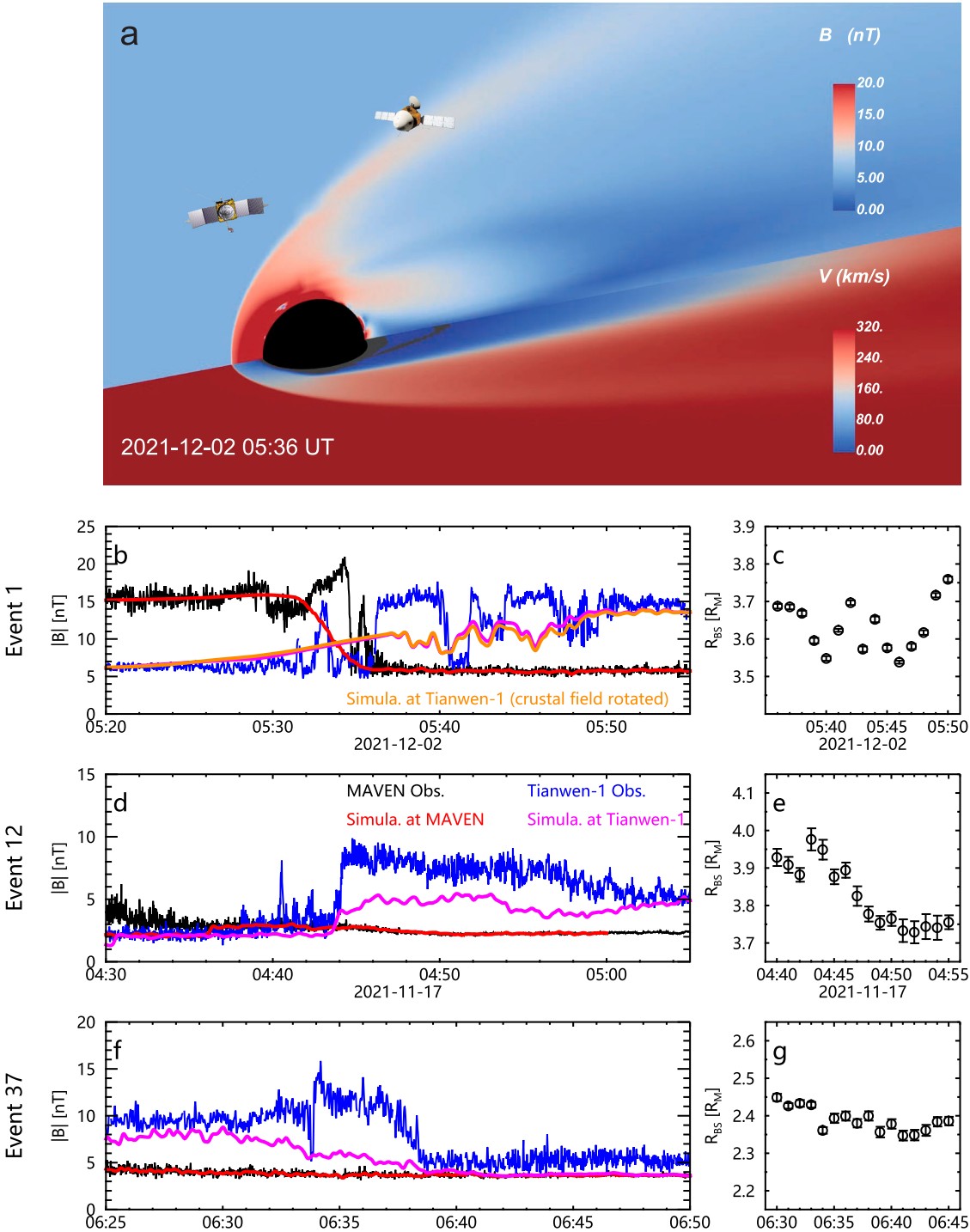

**Fig. 5 | Three-dimensional numerical simulations. a** The magnetic field *(B)* and velocity *(V)* in the *X-Z* plane and *X-Y* plane, respectively, of the Mars Solar Orbital reference frame simulated for event 1. **b** Profiles of the observed magnetic field strength from MAVEN (black lines) and Tianwen-1 (blue lines), and those from the simulated results at the locations of MAVEN (red lines) and Tianwen-1 (magenta lines) of event 1, in which Tianwen-1's trajectory are shifted 15 min earlier to better match the observations, the orange line shows results of the contrast simulation with the strongest crustal magnetism region rotated 90° toward the nightside, with Tianwen-1's trajectroy shifted 30 min earlier to match the observations. **c** locations of the simulated bow shock along the radial direction passing through the vicinity of Tianwen-1 for event 1. The error bars are the 3-sigma uncertainty from the Gaussian fitting to the simulation data (see "Methods", subsection "3D numerical simulations", and Supplementary Fig. 5). **d, e** similar to (**b**) and (**c**), but for event 12, with 7 min shift. **f, g** similar to (**b**) and (**c**), but for event 37. Source data are provided as a Source Data file.

wind speed[23,57]. The gradient at a certain point on the three-dimensional BS surface, *S*,

$$S = Y^2 + Z^2 - (\epsilon^2 - 1)(X - x_F)^2 + 2\epsilon L(X - x_F) - L^2 = 0 \quad (1)$$

can be calculated by

$$\nabla S = \begin{pmatrix} \frac{\partial S}{\partial X} \\ \frac{\partial S}{\partial Y} \\ \frac{\partial S}{\partial Z} \end{pmatrix} = \begin{pmatrix} -2(\epsilon^2 - 1)(X - x_F) + 2\epsilon L \\ 2Y \\ 2Z \end{pmatrix} \quad (2)$$

where $x_F$ is the position of the focus point on the $X$ axis, $\epsilon$ is the conic's eccentricity and $L$ is the semilatus rectum, which could be set as the three parameters from the BS model by Edberg et al.[59]. The model BS normal $\mathbf{n}_{mod}$ is parallel to this gradient,

$$\mathbf{n}_{mod} = \pm \frac{\nabla S}{|\nabla S|} \quad (3)$$

However, it is noteworthy that the location of the BS, $(x, y, z)$, crossed by the spacecraft, may be not on the BS surface depicted by the conic model. Before applying Eqs. 2 and 3 to $(x, y, z)$, we need to scale $L$ and $x_F$ as necessary[57], so that the scaled BS surface passes through the actual BS location $(x, y, z)$. The scaled factor, $\sigma$, can be determined by solving the equation:

$$y^2 + z^2 - (\epsilon^2 - 1)(x - \sigma x_F)^2 + 2\sigma\epsilon L(x - \sigma x_F) - (\sigma L)^2 = 0 \quad (4)$$

Then we have a scaled BS model with the eccentricity $\epsilon$, the focus point $\sigma x_F$, and the semilatus rectum $\sigma L$ and can determine the normal at the BS location $(x, y, z)$ based on Eqs. 2 and 3.

The derived two BS normals, $\mathbf{n}_{MVA}$ and $\mathbf{n}_{mod}$, for the two events are listed in Supplementary Table 3. The difference between them for the two events are both within 6°, suggesting a good consistency. Further, based on the BS normal from the MVA, $\mathbf{n}_{MVA}$, we define the oscillation amplitude of the BS:

$$A_{so} = D\cos\theta \quad (5)$$

where $D$ is the distance between the spacecraft locations at two crossings and $\theta$ is the angle between the spacecraft trajectory and the shock normal.

**Distinguishing solar wind and magnetosheath from MINPA data**

MINPA was designed mainly for the detection of ions and neutral particles in the Martian induced magnetosphere and ionosphere. Here we use its unique orientation to diagnose the solar wind signature. MINPA is a toroidal top-hat electrostatic analyzer followed by a time of flight unit with a base time resolution of 4 s. It provides ion measurements in the energy range 2.8–25.9 keV, with $22.5° \times 5.4°$ angular resolution in a $360° \times 90°$ FOV, and resolves $H^+$, $He^{2+}$, $He^+$, $O^+$, $O_2^+$ and $CO_2^+$. Its location and orientation on the Tianwen−1 orbiter are shown in Supplementary Fig. 4, where $X_b$, $Y_b$ and $Z_b$ show the axes of the orbiter coordinate system. FOV of MINPA ion measurements covers the positive region of the $X$-axis, hence, MINPA measures ions come from the positive region of the $X_b$, which means ions with negative or zero velocity along the $-X_b$ direction. In the solar wind, Tianwen−1's attitude is fixed, with $-Z_b$ pointing to the Sun ($+X_{MSO}$), $Y_b$ pointing to the $Z_{MSO}$, and $X_b$ pointing to the $-Y_{MSO}$. Therefore, in the solar wind, MINPA's FOV covers the negative region of $Y_{MSO}$ axis, which means that MINPA cannot receive the ions coming from the positive region of the $Y_{MSO}$ axis (with negative $Y$ velocity). During the periods of the two cases shown in Fig. 1 and Fig. 3, the solar wind beam was not exactly along $-X_{MSO}$ direction, but slightly deviated toward $-Y_{MSO}$ direction, resulting in a weak solar wind signature. However, when the orbiter entered magnetosheath, the solar wind beam was deflected toward $+Y_{MSO}$ direction, resulting in a clear solar wind signature. The particular orientation of MINPA is helpful to determine if the Tianwen−1 orbiter locates in the solar wind.

**3D numerical simulations**

The BATS-R-US (Block-Adaptive-Tree-Solar wind-Roe-Upwind Scheme)[60] code, a high-performance magnetohydrodynamic (MHD) model that utilizes adaptive mesh refinement (AMR), is used for efficient and accurate simulations. It has been widely used to study

Mars-solar wind interactions and has been validated through comparisons with spacecraft observations[50,61–63]. The global MHD model used in this study is based on the multi-species MHD approach, as described by Ma et al.[50]. The model self-consistently solves plasma interactions between the solar wind, Martian ionosphere, and induced magnetosphere, capturing the effects of crustal field rotation on plasma dynamics.

The computational domain extends from $-24\,R_M$ to $+8\,R_M$ in the $X$-direction, and from $-16\,R_M$ to $+16\,R_M$ in both the $Y$ and $Z$ directions. A non-uniform spherical grid is used, featuring a high radial resolution of 6.4 km near the inner boundary (at an altitude of 100 km). The radial resolution gradually increases to 2000 km near the downstream outer boundary. Throughout the simulation domain, the angular resolution remains constant at 3° in both the longitudinal and azimuthal directions. In total, the simulation domain contains approximately 1.7 million grid cells.

The upstream outer boundary conditions are set using 20-s averaged upstream solar wind parameters observed by MAVEN during the event. A time shift is applied to account for the propagation delay from the upstream outer boundary ($8R_M$) to the MAVEN location. Floating boundary conditions are used for the downstream outer boundary.

At the inner boundary, the $O_2^+$, $O^+$, and $CO_2^+$ densities are set to their photochemical equilibrium values. Reflective boundary condition is applied to the velocity, resulting in near-zero flow velocities at the inner boundary. The plasma temperature is assumed to be twice the corresponding neutral temperature, and the pressure is set accordingly. The magnetic field at the inner boundary is specified to match the crustal magnetic field[64], set to the appropriate solar longitude and latitude for each specific event and rotated with the planet.

The simulations results are displayed in Fig. 5 and a demonstration animation can be found in the Supplementary Movie 1. Due to the limitation of the computational resource on the spatial and temporal resolutions, the BS location in the simulation differs from the observation and the magnetic field fluctuations are not as large as observations. To match the observations, the simulating magnetic field profiles in Fig. 5b are shifted by 15 min (the magenta line) and 30 min (the orange line), respectively. The different shift time is due to the presence/absence of the crustal field, that can alter the position of BS as revealed in previous statistical studies[50]. However, such alterations should be at the time scale of hours as suggested by the simultions[65] rather than the minute-scale investigated here. Similarly, for event 12 and 37, the time shift is 7 min and zero, respectively.

Given that the simulations are MHD-based and thus limited in capturing small-scale or transient features, such as fine BS structures, wave activity, or short-timescale fluctuations, the differences between simulations and observations are expected. Despite these discrepancies, the simulations reproduce the key large-scale characteristics relevant to our study, i.e., the overall amplitude and presence of BS oscillations (Fig. 5 and Supplementary Movie 1) under weakly perturbed solar wind conditions. The general trend and level of oscillatory behavior are consistent with the observations and support our interpretation.

With the simulation results, we identified the BS locations near Tianwen−1 based on the magnetic field along the radial direction that passes through the vicinity of Tianwen-1. Supplementary Fig. 5 provides an example of the magnetic field strength and its gradient at grid points along this radial direction. The BS location is indicated by a sharp change in the magnetic field strength (see Panel a) and an extreme value in the gradient (see Panel b). Due to the limited spatial resolution, we applied a Gaussian fit to the gradient of the magnetic field strength near the BS location (see the red solid line) to determine the peak location, which represents the BS location (see the vertical

dashed line). The 3-sigma of the fitted peak location is treated as the uncertainty.

## Data availability

The MAVEN data is publicly available on the Planetary Data System (https://pds-ppi.igpp.ucla.edu/mission/MAVEN). The Tianwen-1 data is available on the Lunar and Planetary Data Release System (https://moon.bao.ac.cn/web/enmanager/kxsj?missionName=HX1). The MOMAG data displayed in the figures and the numerical simulation data can be retrieved from the official site of the MOMAG team (https://space.ustc.edu.cn/dreams/tw1_momag). Source data of graphs are provided in this paper. Source data are provided with this paper.

## Code availability

The numerical simulation is based on the open sourced BATS-R-US codes (https://github.com/SWMFsoftware).

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

## Acknowledgements

We acknowledge the use of the data from Tianwen-1/MOMAG, Tianwen-1/MIMPA, MAVEN/MAG, and MAVEN/SWIA. The authors from the University of Science and Technology of China are supported by the grants from NSFC (42130204, 42188101, and 42404164) and the Strategic Priority Program of CAS (XDB41000000). L.C. thanks the support of the Excellent Doctoral Overseas Study Program of USTC and the Post-doctoral Fellowship Program of CPSF. Y.W. is particularly grateful for the support of the New Cornerstone Science Foundation through the Xplorer prize.

## Author contributions

Y.W. proposed the analysis of the Martian BS multi-crossing events. L.C. identified the events and made the data analysis. Y.M. carried out the numerical simulations and interpret the results with Y.W. and L.C. Y.W., T.Z., G.W., S.X., Z.Z., Y.C., X.H., Y.L., Z.P., K.L., and L.C. are MOMAG team members and provided the calibrated MOMAG data. A.Z. validated the MINPA data. R.L., J.H., and B.L. validated the MAVEN data and participated in the discussion. L.C. and Y.W. wrote the manuscript together. All authors contributed to shape the manuscript into its final version.

## Competing interests

The authors declare no competing interests.
