## [Transparent Peer Review file · Nature Communications]

Bow Shock Oscillations of Mars Under Weakly Disturbed Solar Wind Conditions

Corresponding Author: Professor Yuming Wang

Version 0:

Reviewer comments:

Reviewer #1

(Remarks to the Author)

The bow shock acts as the first boundary the solar wind encounters as it interacts with a planetary object. The shape and position of the bow shock reflect the conditions of the solar wind and govern the extent of the magnetosheath. While the bow shock position is often steady, rapid changes in the external solar wind conditions such as solar wind dynamic pressure, IMF strength and orientation, and magnetosonic Mach number can generate oscillations in bow shock position.

This study investigates two interesting cases in which the bow shock shows signs of oscillation despite the upstream solar wind conditions remaining steady. This analysis was conducted using simultaneous measurements of two spacecraft: Tianwen-1 and MAVEN.

During these events, MAVEN was positioned in the solar wind closer to the subsolar point, and Tianwen-1 was positioned near the bow shock boundary. MAVEN measured steady solar wind conditions (i.e., no significant changes in dynamic pressure, IMF orientation, IMF magnitude, or magnetosonic Mach number). Despite these steady conditions, Tianwen-1 measured bow shock oscillations as the bow shock passed over the spacecraft multiple times within a few tens of minutes. The authors therefore suggest that these bow shock oscillations are attributable to internal dynamics, possibly involving Mars' crustal magnetic fields, rather than external phenomena driven by changes in solar wind.

This is a compelling study that utilizes the dual-spacecraft analysis capabilities of Tianwen-1 and MAVEN. The real-time comparison between solar wind conditions and bow shock boundary locations at Mars is novel and compelling. These two case studies have interesting implications for the solar wind interaction at Mars. However, more work needs to be done to demonstrate that the bow shock oscillations are not driven by the solar wind transient structures discussed in the paper. Furthermore, a more in-depth discussion about the role of the strong crustal magnetic anomalies is warranted and would increase the impact of the work.

The suggestions given below should be considered a major revision.

Line 51: The citation listed (citation 14) focuses on simulations of magnetotail reconnection at Mars, which occurs between the lobes of the induced magnetotail. I would suggest adding citations such as Harada et al., 2018, 2020, which are data driven studies that focus on reconnection between crustal fields and the draped IMF.

Line 60-63: Could you give some examples of internally-driven bow shock oscillations at Earth? Why exactly would we expect to see something different at Mars?

Figure 1: I'm curious about the jump in magnetic field strength at ~05:33, before the dashed line (1). Is this also a brief encounter with the bow shock? The plasma data also seem to agree. Please discuss.

Figure 1 and 3: Both figures would benefit from more time markers on the horizontal axis. It's helpful when you refer to specific minutes to cross reference to the corresponding place on the figure.

Lines 91-107: The conclusions of this study rest on the argument presented in this and the following paragraphs and therefore must be expanded on further. The extended Figure 3 does a good job of highlighting the magnetic field and plasma signatures for these transient solar wind structures, but this figure is lacking a vertical and horizontal axis. What is the energy

range of the plasma for these events? What about the temporal scale?

This begs the question: what are the energy and time scales of these transient events? Are they comparable with Tianwen-1's observing cadence and energy range? Is there any evidence of Tianwen-1 observing these transient events already? Please include a couple paragraphs discussing the plasma sampling specifics of Tianwen-1 to create a more convincing argument that had these oscillations been caused by any of the transient events listed, Tianwen-1 would have been able to observe them.

Within this context, the schematic shown in Figure 2 is very helpful. Could you discuss the orientation of the IMF with respect to the bow shock for the two case studies you present? How does this compare to the schematic? Was Tianwen-1 sampling a quasi-parallel bow shock, or a quasi-perpendicular bow shock? This could also help with the argument against some of the transient events often associated with the quasi-parallel shock.

Lines 108-121: The distinction between a rippling bow shock and an oscillating bow shock can be made via minimum variance analysis (MVA). This analysis is quite similar to DiBraccio et al., 2017, which distinguished "kink-like" current sheet flapping from "steady-flapping" using MVA using MAVEN data..

You claim that MVA cannot be used to determine the bow shock normal due to the short duration between boundary crossings. For both cases presented here, there are roughly 2-3 minutes at least between crossings. However, the flapping current sheet study cited above performed MVA on multiple current sheet crossings that occur within 1.5 minutes of one another. This suggests that MVA can be useful to compare bow shock normal directions for the crossings presented here.

Please include a discussion of the comparison between individual bow shock normal directions determined by MVA to distinguish between a rippled bow shock and a bow shock that is undergoing internally driven oscillations.

Lines 156-163: Internally driven oscillations at bow shocks of other planets is a necessary discussion point for this study. However, more discussion should be included about internal drivers to bow shock oscillations at Earth. The final paragraph suggests that crustal magnetic field reconnection may be playing a role in BS oscillations at Mars. Is there evidence for reconnection causing bow shock oscillations at Earth's magnetosphere? If not, why should we expect reconnection to play a role at Mars?

Lines 175-177: I'm confused by this. It seems like from Figure 4 there may be a preferred crustal field location with respect to the subsolar point for these oscillations to occur (i.e., when the strongest crustal fields are on the duskside). The fact that single crossing events cover a wide variety of longitudes seems to support the crustal field influence idea, not disprove it.

Lines 178-180: This is an interesting claim and is worth exploring more. Schneider et al., 2021, Xu et al., 2022, Johnston et al., 2023 and Bowers et al., 2023 found that auroral signatures over the strongest crustal fields (centered at 180 deg east longitude and 45 deg south latitude) are favored when the strongest crustal fields are on the duskside and the upstream IMF is in the -BY orientation. Johnston et al., 2023 and Bowers et al., 2023 argued that this was due to enhanced reconnection.

It seems like your two case studies may also be in the conditions ideal for reconnection with this strongest crustal field region (IMF -BY conditions, and strongest crustal fields on the duskside).

This is worth a discussion to go along with the speculative reconnection argument you outline here. Also, a sentence or two is required to argue why reconnection may be playing a role in bow shock oscillations. Does it have to do the crustal magnetic field changing connectivity due to reconnection?

References:

- Harada, Y., Halekas, J. S., DiBraccio, G. A., Xu, S., Espley, J., McFadden, J. P., et al. (2018). Magnetic reconnection on dayside crustal magnetic fields at Mars: MAVEN observations. *Geophysical Research Letters*, 45, 4550–4558. <https://doi.org/10.1002/2018GL077281>
- Harada, Y., Halekas, J. S., Xu, S., DiBraccio, G. A., Ruhunusiri, S., Hara, T., et al. (2020). Ion jets within current sheets in the Martian magnetosphere. *Journal of Geophysical Research: Space Physics*, 125, e2020JA028576. <https://doi.org/10.1029/2020JA028576>
- DiBraccio, G. A., et al. (2017), MAVEN observations of tail current sheet flapping at Mars, *J. Geophys. Res. Space Physics*, 122, 4308–4324, doi:10.1002/2016JA023488.
- Schneider, N. M., Milby, Z., Jain, S. K., Gérard, J.-C., Soret, L., Brain, D. A., et al. (2021). Discrete aurora on Mars: Insights into their distribution and activity from MAVEN/IUVS observations. *Journal of Geophysical Research: Space Physics*, 126, e2021JA029428. <https://doi.org/10.1029/2021JA029428>
- Xu, S., Mitchell, D. L., McFadden, J. P., Schneider, N. M., Milby, Z., Jain, S., et al. (2022). Empirically determined auroral electron events at Mars—MAVEN observations. *Geophysical Research Letters*, 49, e2022GL097757. <https://doi.org/10.1029/2022GL097757>

Johnston, B. J., Schneider, N. M., Jain, S. K., Milby, Z., Deighan, J., Bowers, C. F., et al. (2023). Discrete aurora at Mars: Insights into the role of magnetic reconnection. *Geophysical Research Letters*, 50, e2023GL104198. <https://doi.org/10.1029/2023GL104198>

Bowers, C. F., DiBraccio, G. A., Slavin, J. A., Johnston, B., Schneider, N. M., Brain, D. A., & Azari, A. (2023). Evidence for magnetic reconnection as the precursor to discrete aurora at Mars. *Journal of Geophysical Research: Space Physics*, 128, e2023JA031622. <https://doi.org/10.1029/2023JA031622>

Reviewer #2

(Remarks to the Author)

Review of Internally Driven Oscillations of Martian Bow Shock by Cheng et al.

This paper focuses on bow shock variability in the flank of Mars during solar wind steady conditions and the potential for internal drivers to be the cause for this variability. The work is based on simultaneous observations of Tianwen-1 and MAVEN.

While the topic is very interesting and I fully agree with the authors that there is much to be investigated in this respect, I do not think this paper show evidence for those internal driven oscillations that authors claim. The analysis of the driver mechanisms should be extended and claims validated with other observations, as at the moment, everything is based on speculations. It is true that the paper has a good description of potential sources of variability for the bow shock, but they are not checked or validated, and many others are missing, such as Kelvin-Helmholtz instabilities, or simply magnetic reconnection in the tail (as the data shown in this paper come from the flank area). Moreover, the actual solar wind is not investigated (i.e., in which part of the parker spiral polarity these observations occurred? were they during the steady part of a slow or fast solar wind stream? etc.). Also, multiple bow shock crossing observations have been extensively report before, even in the literature cited by this paper.

Therefore, for the above reasons, I do not think this paper is ready for publication, and my suggestion is to focus on characterising those internal drivers with observations or modelling, as that would be a much appreciated finding, but unfortunately at the moment, I do not think this paper has enough maturity.

Version 1:

Reviewer comments:

Reviewer #1

(Remarks to the Author)

Thank you for addressing many of the concerns that were raised in the previous review. The inclusion of MVA along the boundary and a discussion of the upstream IMF with respect to the shock and regarding magnetic reconnection significantly improve the paper.

I completely agree with the authors that this is a compelling study that is worth further investigation. The two events presented here are indeed quite interesting and I agree with the authors that they show evidence for internal dynamics driving bow shock oscillations.

However, the explanations for the observed oscillations and the limited number of events (only two) make the conclusions of this manuscript too speculative for publication in its current form. In my previous review, I asked whether there is evidence of reconnection affecting bow shocks at other planets, such as Earth. Such evidence would strongly support the argument that similar processes might occur at Mars and would reduce the level of speculation required to substantiate the conclusions presented here. It appears that such evidence is lacking, which makes the current argument more speculative.

I recently came across Garnier et al., 2022 that discusses the influence of crustal fields on the Martian bow shock location. It is possible that the crustal field-reconnection argument could be strengthened with references to the conclusions presented in that manuscript. If the authors agree, I would encourage them to resubmit one more time with a more robust argument that reconnection could be driving bow shock oscillations. In its present form, the argument is too speculative to be published here.

Figure 3 and Lines 156-164: I am confused by panel g. We see a big change in magnetosonic mach number, which you explain is part of the SWIA mode switch. Magnetosonic mach numbers < 2 in the solar wind at Mars' orbit would be extremely rare, which makes me doubt the accuracy of this prediction. Can you explain more why we may see the jump in Mach number and how accurate these predictions are?

Line 180: I recently came across Garnier et al., 2022 that investigated the influence of Mars crustal fields on bow shock location. This investigation has important implications for your work, I would strongly recommend that you cite this paper and discuss its implications here.

Line 187-191: It seems to me that the presence of single BS crossings under steady SW conditions near longitudes of the multiple crossing events (i.e. black triangles near red circles in Figure 4) suggests that the BS is not always variable, event under similar subsolar longitude conditions. Is this the correct interpretation? To me, this suggests that it may be some interaction between the crustal fields and upstream conditions that leads to variable BS events. Am I understanding correctly?

References:

Garnier P., Jaquey C., Gendre X., Génot V., Mazelle C., et al., (2022). The Influence of Crustal Magnetic Fields on the Martian Bow Shock Location: A Statistical Analysis of MAVEN and Mars Express Observations. *Journal of Geophysical Research: Space Physics*. <https://doi.org/10.1029/2021JA030146>

Reviewer #2

(Remarks to the Author)

Second review of Internally Driven Oscillations of Martian Bow Shock by Cheng et al.

This paper focuses on bow shock variability in the flank of Mars during solar wind steady conditions and the potential for internal drivers to be the cause for this variability. The work is based on simultaneous observations of Tianwen-1 and MAVEN.

I appreciate very much the authors dedication with this work, which topic is very important for our understanding of Martian magnetospheric dynamics. However, my review is still the same as it was the first time. I do not think there is enough evidence in this paper to prove the authors' claim "it is the first report of the unexpected and new phenomenon that may stimulate follow-up studies for the mechanisms behind the phenomenon.". There is a long discussion in the paper about "speculative" arguments (this word even appears twice!) and non of them based on an analysis of the physics behind the oscillations. There is also a Table with plots from other papers that are out of context as we do not know the conditions where those observations were taken, or if they are comparable to the observations of this work.

Moreover, there is not plot that shows the trajectories of both Tianwen and MAVEN. To me they look like at least MAVEN is skimming the bow shock, but it is difficult to get a robust conclusion without that basic information. This is the main reason that one can speculate with theories, but they do not demonstrate, in my opinion, that that they are the real reason behind the speculation.

I would like to insist that the topic is very interesting, and certainly I would be in favour of giving a more positive review if authors can prove their claims based on observations, theoretical analysis or numerical simulations that what they speculate is the real reason. Also, I would appreciate if replies would be included in the paper and not only to this referee, as other readers may have the same concerns.

Version 2:

Reviewer comments:

Reviewer #1

(Remarks to the Author)

The latest version of manuscript is a significant change to the previous versions, focusing instead on the impact of magnetosonic Mach number of the susceptibility of the bow shock. They argue that lower upstream Mach numbers cause large-scale oscillations of the Martian bow shock, which could explain why Tianwen-1 observed multiple bow shock crossings while MAVEN measured relatively steady upstream conditions. The inclusion of BATS-R-US simulations of the bow shock are a welcome inclusion to the study, and show promise to support these conclusions.

However, I still find that the conclusions of this study are not adequately supported by the results as they are presented here. The multi-crossing events do seem to take place under low Mach numbers than the single crossing events, but the difference between these two values (4.7 vs. 5.0) is not very significant. It is still possible that this small change in Mach number could explain the results, but this is not demonstrated in the study as it is written currently. To show that this small difference in Mach numbers leads to a different number of crossings, one would need to run two simulations—one with a Mach number of 4.7 and another with 5.0—and demonstrate that this alters the bow shock oscillation behavior observed by Tianwen-1.

Furthermore, the simulations appear promising in illustrating the differences in bow shock oscillations between Tianwen-1 and MAVEN, but these differences are difficult to discern in their current presentation. The movies would be much clearer if they displayed the shock position flattened along the horizontal axis, highlighting how the boundary motion varies at different locations along the shock—similar to panels d and e in Figure 5.

Including a single crossing event (Event 12) as a comparison to the multi-crossing events is a good idea. However, additional details about the MHD simulation for Event 12 are needed to substantiate the study's conclusions. Specifically, the authors should include a time series—comparable to Figures 5b and 5c—showing both the observations and simulation results for Event 12, in order to clearly demonstrate that one scenario leads to multiple crossings at Tianwen-1 while the

other results in only a single crossing.

While I still agree that these events show promise for illuminating some interesting bow shock physics, the conclusions presented here are still not adequately supported by the results.

Lines 111: The bow shock also heats plasma in the magnetosheath compared to the solar wind. This would also broaden the distribution of protons and increase counts in the limited FOV.

Line 193: Please clarify what “angular angle” means.

Lines 229-230: The difference between a Mach number of 4.7 vs. a Mach number of 5.0 is not very significant. In order to demonstrate that this small difference in Mach numbers result in a different number of crossings, one would need to produce two simulations (one with Mach number of 4.7 and another with Mach number of 5.0) and demonstrate that this would change bow shock oscillation behaviour at Tianwen-1.

Line 245-246: This choice to shift the simulation results by 15 and 30 minutes should be justified and discussed in greater detail. The similarities between Case 1 and Case 2 (Figure 5b and 5c) is somewhat misleading because the different shifts of the model results suggest the bow shock does behave differently given different crustal field locations.

Figure 4b: It is difficult to distinguish the diamonds from the circles. I would suggest making this difference between SWIA modes clearer. Maybe circles and triangles?

Lines 256-259: I believe that the choice for Event 12 as a useful comparison requires more justification. Is Event 12 also a quasi-perpendicular shock? Does this conclusion hold for other events with similar Mach numbers?

Line 260: The difference between Movies 1 and 2 is very difficult to see at this scale. The format of these movies should be changed to better demonstrate the point the authors are trying to make.

Figure 5: It is necessary to compare this with the simulated data from the single crossing event (Event 12) to demonstrate that larger Mach numbers suppress these oscillations. It is important to include another time series (similar to 5b and 5c) but for the observations and simulated results of Event 12.

Supplementary Movies 1 and 2: The simulations show promise for demonstrating the difference in oscillation of the bow shock at Tianwen 1 vs. MAVEN, but this is quite difficult to see in its current form. These movies would greatly benefit from showing the shock position flattened out on the horizontal axis and showing how this boundary motion is different at different points along the shock (similar to panels d and e in Figure 5).

Lines 266-268: Mach number doesn't only depend on the speed of the solar wind. Based on your argument, would large IMF $|B|$ also lead to bow shock oscillations because Mach number is inversely proportional to $|B|$? What about solar wind density?

Reviewer #2

(Remarks to the Author)

Review of Oscillations of Mars' Bow Shock Under Weakly Disturbed Solar Wind Conditions by Cheng et al.

This paper focuses on bow shock variability in the flank of Mars during steady conditions and the potential for internal drivers to be the cause for this variability. The work is based on simultaneous observations of Tianwen-1 and MAVEN and simulations.

The work has significantly improved since last version. The new simulation adds a great value and make the conclusions much more evident than in the previous versions. The discussion is also good and appropriate.

I only have a few minor suggestions in this round:

- In the abstract and in other parts of the text: I don't think it is appropriate to say “the Mach number is probably the most sensitive...” the word “probably” reduces credibility to the results.

- The 35 crossings that are compared to the two selected, how they were chosen? From Table 4 I understand they come from similar periods of time, but this is not very obvious to infer from the paper. Can authors add more clarifications on this aspect in Section “cause of the oscillations”? Are all those cases comparable in terms of solar wind activity, IMF direction, Mach number, Tianwen-1 and MAVEN locations, latitude, etc?

- Figure 4 is a bit confusing. Which ones are the crossings of Figures 2 and 3? Not obvious. Also, I see at least 8 crossings with similar low Mach number and many others with similar “higher” Mach number. How the authors conclude that this parameter is a key to characterise large-scale disturbances? The plot is not obvious to interpret, and therefore, the conclusions are difficult to take. Have all these 35 events been analyse in order to characterise the flapping of the

magnetosphere?

- I still don't see the trajectories of both MAVEN and Tianwen-1 in Figures 1 and 3, and the interpretation of the results is difficult. I only see the transit of the data that have been plotted. Ideally, it would be good to have the whole orbit plotted.

- Are the numerical simulations appropriate? The pink profiles in Figure 5 show some discrepancies with Tianwen-1 blue profiles, and not sure it really captures the "basic observational features" as where the data has a dip, the simulation has an increase, an viceversa. Are these discrepancies expected and acknowledged? It deserves more explanations in the paper.

- Table 1: same comment as in the previous revision. What's the point of this table when the conditions are not comparable? At least I would remove the plots within the table as they only introduce confusion and keep the last 3 rows, and please explain all the acronyms in that figure, like for example, what is SLAMS? Or HFA/SHF? Although it is said in the text, I believe the table should be self-explanatory.

Version 3:

Reviewer comments:

Reviewer #1

(Remarks to the Author)

The latest version of this manuscript sufficiently addresses my concerns from the previous version. The inclusion of multiple simulations for single crossing events strengthens the study, supporting the claim that lower Mach number shocks may be more prone to oscillations compared to high Mach number shocks on the flanks.

I am satisfied with these revisions and am happy to recommend this study for publication.

Reviewer #2

(Remarks to the Author)

The authors have satisfactorily addressed all my previous concerns, and I am pleased to recommend the paper for publication.

Dear Editor and Reviewers,

Thank you for your thorough review and valuable feedback, which have greatly contributed to
improving the quality of our paper. We have addressed each comment in detail and revised the
manuscript accordingly. Please find our detailed responses below, highlighted in blue.
Corresponding changes in the manuscript are also marked in blue for your convenience.

Reviewer #1 (Remarks to the Author):

The bow shock acts as the first boundary the solar wind encounters as it interacts with a planetary
object. The shape and position of the bow shock reflect the conditions of the solar wind and govern
the extent of the magnetosheath. While the bow shock position is often steady, rapid changes in the
external solar wind conditions such as solar wind dynamic pressure, IMF strength and orientation,
and magnetosonic Mach number can generate oscillations in bow shock position.

This study investigates two interesting cases in which the bow shock shows signs of oscillation
despite the upstream solar wind conditions remaining steady. This analysis was conducted using
simultaneous measurements of two spacecraft: Tianwen-1 and MAVEN.

During these events, MAVEN was positioned in the solar wind closer to the subsolar point, and
Tianwen-1 was positioned near the bow shock boundary. MAVEN measured steady solar wind
conditions (i.e., no significant changes in dynamic pressure, IMF orientation, IMF magnitude, or
magnetosonic Mach number). Despite these steady conditions, Tianwen-1 measured bow shock
oscillations as the bow shock passed over the spacecraft multiple times within a few tens of minutes.
The authors therefore suggest that these bow shock oscillations are attributable to internal dynamics,
possibly involving Mars' crustal magnetic fields, rather than external phenomena driven by changes
solar wind.

This is a compelling study that utilizes the dual-spacecraft analysis capabilities of Tianwen-1 and
MAVEN. The real-time comparison between solar wind conditions and bow shock boundary
locations at Mars is novel and compelling. These two case studies have interesting implications for
the solar wind interaction at Mars. However, more work needs to be done to demonstrate that the
bow shock oscillations are not driven by the solar wind transient structures discussed in the paper.
Furthermore, a more in-depth discussion about the role of the strong crustal magnetic anomalies is
warranted and would increase the impact of the work.

The suggestions given below should be considered a major revision.

Thank you for your positive and constructive comments and suggestions. We have tried to address
all the comments and hope the revised version is satisfactory.

Line 51: The citation listed (citation 14) focuses on simulations of magnetotail reconnection at Mars,
which occurs between the lobes of the induced magnetotail. I would suggest adding citations such
as Harada et al., 2018, 2020, which are data driven studies that focus on reconnection between
crustal fields and the draped IMF.

Reply: Thank you for the suggestion. We add these two citations in the Introduction.

Line 60-63: Could you give some examples of internally-driven bow shock oscillations at Earth?
Why exactly would we expect to see something different at Mars?

Reply: To our knowledge, no internally-driven bow shock oscillation was reported at Earth. Please
let us know if we missed something. The sentences there just argue that the dynamics of the bow
shock at Mars is worth to be studied though that at Earth has been extensively studied with multiple
spacecraft observations, because Earth has a global and strong magnetic field, while Mars displays
only localized crustal fields.

Figure 1: I'm curious about the jump in magnetic field strength at ~05:33, before the dashed line
(1). Is this also a brief encounter with the bow shock? The plasma data also seem to agree. Please
discuss.

Reply: Thank you for the suggestion. The jump in $|B|$ at ~05:33 may indicate a brief encounter with
the bow shock, as $|B|$ and the proton flux increased to the level of the sheath, but decreased to the
level of the solar wind soon. We mention it in the revised manuscript (Lines 87-90).

Figure 1 and 3: Both figures would benefit from more time markers on the horizontal axis. It's
helpful when you refer to specific minutes to cross reference to the corresponding place on the figure.

Reply: Thank you for the suggestion. We add additional time markers in Figures 1 and 3.

Lines 91-107: The conclusions of this study rests on the argument presented in this and the following
paragraphs and therefore must be expanded on further. The extended Figure 3 does a good job of
highlighting the magnetic field and plasma signatures for these transient solar wind structures, but
this figure is lacking a vertical and horizontal axis. What is the energy range of the plasma for these
events? What about the temporal scale?

Reply: Thank you for the suggestion. We add the energy range of these transient events in Extended
Data Figure 3 (now Table 1 in the Main Text). The horizontal time axes of the magnetic field and
ion flux figures are identical in each event. The durations of each type of event vary from case to
case; therefore, we include the typical durations of each type of event at Earth and Mars, respectively.
References for the figures and event durations at Earth and Mars are provided in the table.

	Foreshock						Sheath	
	HFAs/ SHFAs	Foreshock Bubbles	Foreshock Cavities/ Foreshock Cavitons	Foreshock Compressional Boundary	Density Holes	SLAMS	Sheath Jets	Magnetic Holes
B-field								Ion Flux								Fig. Source	[56]	[57]	[58]		[59]	[60]	[61]	[62]
Duration (Earth)³⁵	Minutes	Minutes	Minutes	Minutes	Seconds	~10 s	Minutes	Milliseconds to Minutes
Duration (Mars)	~1 Minute ^{56,63}	~1 Minute ⁵⁷	Minutes ⁵⁸	Seconds ⁵⁸			Minutes ⁶⁴	

This begs the question: what are the energy and time scales of these transient events? Are they
comparable with Tianwen-1's observing cadence and energy range? Is there any evidence of
Tianwen-1 observing these transient events already? Please include a couple paragraphs discussing
the plasma sampling specifics of Tianwen-1 to create a more convincing argument that had these
oscillations been caused by any of the transient events listed, Tianwen-1 would have been able to
observe them.

Reply: Thank you for the suggestions.

Variations in the ion flux caused by shock transients can be seen in energies near 1 keV, at time
scales of minutes (mentioned in Lines 99-102), which are within the plasma specifics of MINPA
(mentioned in Lines 535-538 in Methods).

As to the two events studied in this work, the rapid changes in the proton fluxes, as well as the
magnetic field, between the levels of the upstream solar wind and the magnetosheath distinguish
shock motions from shock transient structures (mentioned in Lines 104-114).

Within this context, the schematic shown in Figure 2 is very helpful. Could you discuss the
orientation of the IMF with respect to the bow shock for the two case studies you present? How
does this compare to the schematic? Was Tianwen-1 sampling a quasi-parallel bow shock, or a quasi-
perpendicular bow shock? This could also help with the argument against some of the transient
events often associated with the quasi-parallel shock.

Reply: Thank you for the suggestion. As listed in Extended Data Table 3, the angle between the
shock normal and the IMF are 70° and 87°, respectively. Hence, both events are classified as quasi-
perpendicular shocks, further excluding the likelihood of transient events that primarily occur near
quasi-parallel shocks. We mentioned it in Lines 127-131.

Lines 108-121: The distinction between a rippling bow shock and an oscillating bow shock can be
made via minimum variance analysis (MVA). This analysis is quite similar to DiBraccio et al., 2017,

which distinguished “kink-like” current sheet flapping from “steady-flapping” using MVA using
 MAVEN data..

You claim that MVA cannot be used to determine the bow shock normal due to the short duration
 between boundary crossings. For both cases presented here, there are roughly 2-3 minutes at least
 between crossings. However, the flapping current sheet study cited above performed MVA on
 multiple current sheet crossings that occur within 1.5 minutes of one another. This suggests that
 MVA can be useful to compare bow shock normal directions for the crossings presented here.

Please include a discussion of the comparison between individual bow shock normal directions
 determined by MVA to distinguish between a rippled bow shock and a bow shock that is undergoing
 internally driven oscillations.

Reply: Thank you for the suggestion. We calculate the shock normal of each crossing in each case
 and check the intersection angles between individual shock normal directions. The tables below
 have been added to the manuscript as Extended Data Table 1 and 2. As one can see, the shock normal
 direction changed little at each crossing, which indicates the oscillation of a planar shock. Relevant
 discussion is added in Lines 119-126.

Parameters of shock normal of the event on 2021 December 2. The normal are calculated by the
 minimum variance analysis, using the magnetic field data within the listed time window. \hat{n}
 represents the normal in the entire event, while \hat{n}_1 , \hat{n}_2 and \hat{n}_3 represent the normal of each
 crossing.

	Time Window	Value	Intersection Angle			
			\hat{n}_1	\hat{n}_2	\hat{n}_3	\hat{n}
\hat{n}_1	05:34:00 ~ 05:38:00	[0.56, 0.17, -0.81]		6.3°	5.0°	2.2°
\hat{n}_2	05:38:00 ~ 05:41:00	[0.60, 0.07, -0.79]	6.3°		1.4°	8.4°
\hat{n}_3	05:41:00 ~ 05:42:15	[0.60, 0.10, -0.79]	5.0°	1.4°		7.1°
\hat{n}	05:25:00 ~ 05:50:00	[0.53, 0.20, -0.82]	2.2°	8.4°	7.1°	

Parameters of shock normal of the event on 2021 December 25.

	Time Window	Value	Intersection Angle			
			\hat{n}_1	\hat{n}_2	\hat{n}_3	\hat{n}
\hat{n}_1	20:23:00 ~ 20:28:00	[0.58, 0.34, -0.74]		4.1°	1.6°	1.2°
\hat{n}_2	20:28:00 ~ 20:32:30	[0.58, 0.40, -0.71]	4.1°		4.1°	3.3°
\hat{n}_3	20:32:30 ~ 20:36:00	[0.56, 0.35, -0.75]	1.6°	4.1°		2.4°

\hat{n}	20:15:00 ~ 20:40:00	[0.59, 0.34, -0.73]	1.2°	3.3°	2.4°	
-----------	---------------------	---------------------	------	------	------	--

Lines 156-163: Internally driven oscillations at bow shocks of other planets is a necessary discussion
 point for this study. However, more discussion should be included about internal drivers to bow
 shock oscillations at Earth. The final paragraph suggests that crustal magnetic field reconnection
 may be playing a role in BS oscillations at Mars. Is there evidence for reconnection causing bow
 shock oscillations at Earth's magnetosphere? If not, why should we expect reconnection to play a
 role at Mars?

Reply: Thank you for your insightful comment regarding the need for a discussion on internally
 driven oscillations at bow shocks of Earth. Additional discussions are added in Lines 194-199.

At Earth, while reconnection is a well-known driver of various magnetospheric processes, there is
 no direct evidence linking magnetic reconnection to bow shock oscillations specifically. However,
 the Martian environment differs from Earth and other planets, particularly due to Mars' lack of a
 global magnetic field and the presence of highly strong, localized crustal magnetic fields. These
 crustal fields at Mars can interact with the IMF in a way that may be more conducive to triggering
 reconnection events that could influence the bow shock.

The unique configuration at Mars—where localized crustal fields are directly exposed to the solar
 wind—could create conditions that are more favorable for such reconnection-driven processes,
 potentially leading to bow shock oscillations. This is why we propose that, although similar
 processes might not be as prominent at Earth, the specific conditions at Mars could allow for
 reconnection to play a more significant role in driving bow shock oscillations.

Lines 175-177: I'm confused by this. It seems like from Figure 4 there may be a preferred crustal
 field location with respect to the subsolar point for these oscillations to occur (i.e., when the
 strongest crustal fields are on the duskside). The fact that single crossing events cover a wide variety
 of longitudes seems to support the crustal field influence idea, not disprove it.

Reply: Sorry for the ambiguous expression. We do claim that there may be a preferred duskside
 location of the crustal field for shock oscillations during steady solar wind. However, single crossing
 events may also occur at the duskside location. Hence, we suggest that the crustal field may be not
 the only cause for shock oscillations during steady solar wind.

We examined the IMF conditions during the events, of which the subsolar longitude is within 70-
 150 degrees (or the strongest crustal field region is located on the duskside). The figure above
 illustrates the distributions of subsolar longitudes and IMF clock angles. The IMF clock angle is
 calculated as $\text{acos}(B_y/B_z)$, with positive/negative values indicating $+B_y/-B_y$ orientations. Similar to
 Figure 4, the red circles denote BS oscillation events under the steady solar wind conditions, while
 the black triangles indicate single-crossing events. It can be observed that for the two steady-
 oscillation events, the IMF had a $-B_y$ orientation. However, there were also three single-crossing
 events with a duskside-located strongest crustal field region and $-B_y$ IMF orientation. This result
 may lead to several possibilities. One is that the longitude range to favorite the crustal-field related
 BS oscillation is a relatively narrow range from about 100 to 130 degrees. Another one is that an
 oscillating BS is not always observable by the spacecraft. For example, if the BS oscillates behind
 the spacecraft's initial crossing point, the spacecraft would only cross the BS once. Besides, it is
 also possible that there is other unknown causes for such an unexpected phenomenon.

The above discussion has been added in the manuscript in Lines 210-222.

Lines 178-180: This is an interesting claim and is worth exploring more. Schneider et al., 2021, Xu
 et al., 2022, Johnston et al., 2023 and Bowers et al., 2023 found that auroral signatures over the
 strongest crustal fields (centered at 180 deg east longitude and 45 deg south latitude) are favored
 when the strongest crustal fields are on the duskside and the upstream IMF is in the $-BY$ orientation.
 Johnston et al., 2023 and Bowers et al., 2023 argued that this was due to enhanced reconnection.

It seems like your two case studies may also be in the conditions ideal for reconnection with this
 strongest crustal field region (IMF $-BY$ conditions, and strongest crustal fields on the duskside).

This is worth a discussion to go along with the speculative reconnection argument you outline here.
 Also, a sentence or two is required to argue why reconnection may be playing a role in bow shock
 oscillations. Does it have to do the crustal magnetic field changing connectivity due to reconnection?

Reply: Thank you for the insightful comments. We add some discussion in the last section in the

Main Text.

References:

Harada, Y., Halekas, J. S., DiBraccio, G. A., Xu, S., Espley, J., McFadden, J. P., et al. (2018).
Magnetic reconnection on dayside crustal magnetic fields at Mars: MAVEN observations.
*Geophysical Research Letters*, 45, 4550–4558. <https://doi.org/10.1002/2018GL077281>

Harada, Y., Halekas, J. S., Xu, S., DiBraccio, G. A., Ruhunusiri, S., Hara, T., et al. (2020). Ion jets
within current sheets in the Martian magnetosphere. *Journal of Geophysical Research: Space*
*Physics*, 125, e2020JA028576. <https://doi.org/10.1029/2020JA028576>

DiBraccio, G. A., et al. (2017), MAVEN observations of tail current sheet flapping at Mars, *J.*
*Geophys. Res. Space Physics*, 122, 4308–4324, doi:10.1002/2016JA023488.

Schneider, N. M., Milby, Z., Jain, S. K., Gérard, J.-C., Soret, L., Brain, D. A., et al. (2021). Discrete
aurora on Mars: Insights into their distribution and activity from MAVEN/IUVS observations.
*Journal of Geophysical Research: Space Physics*, 126, e2021JA029428.
<https://doi.org/10.1029/2021JA029428>

Xu, S., Mitchell, D. L., McFadden, J. P., Schneider, N. M., Milby, Z., Jain, S., et al. (2022).
Empirically determined auroral electron events at Mars—MAVEN observations. *Geophysical*
*Research Letters*, 49, e2022GL097757. <https://doi.org/10.1029/2022GL097757>

Johnston, B. J., Schneider, N. M., Jain, S. K., Milby, Z., Deighan, J., Bowers, C. F., et al. (2023).
Discrete aurora at Mars: Insights into the role of magnetic reconnection. *Geophysical Research*
*Letters*, 50, e2023GL104198. <https://doi.org/10.1029/2023GL104198>

Bowers, C. F., DiBraccio, G. A., Slavin, J. A., Johnston, B., Schneider, N. M., Brain, D. A., & Azari,
207 A. (2023). Evidence for magnetic reconnection as the precursor to discrete aurora at Mars.
*Journal of Geophysical Research: Space Physics*, 128, e2023JA031622.
<https://doi.org/10.1029/2023JA031622>

Reviewer #2 (Remarks to the Author):

Review of Internally Driven Oscillations of Martian Bow Shock by Cheng et al.

This paper focuses on bow shock variability in the flank of Mars during solar wind steady conditions
and the potential for internal drivers to be the cause for this variability. The work is based on
simultaneous observations of Tianwen-1 and MAVEN.

While the topic is very interesting and I fully agree with the authors that there is much to be
investigated in this respect, I do not think this paper show evidence for those internal driven
oscillations that authors claim. The analysis of the driver mechanisms should be extended and claims
validated with other observations, as at the moment, everything is based on speculations. It is true
that the paper has a good description of potential sources of variability for the bow shock, but they
are not checked or validated, and many others are missing, such as Kelvin-Helmholtz instabilities,
or simply magnetic reconnection in the tail (as the data shown in this paper come from the flank
area). Moreover, the actual solar wind is not investigated (i.e., in which part of the parker spiral
polarity these observations occurred? were they during the steady part of a slow or fast solar wind
stream? etc.). Also, multiple bow shock crossing observations have been extensively report before,
even in the literature cited by this paper.

Therefore, for the above reasons, I do not think this paper is ready for publication, and my suggestion
is to focus on characterising those internal drivers with observations or modelling, as that would be
a much appreciated finding, but unfortunately at the moment, I do not think this paper has enough
maturity.

Reply:

Thank you for the comments. We have tried to address all your concerns. Please see the revised
manuscript and replies below. Hope this version is satisfactory.

(1) Kelvin-Helmholtz instabilities

Firstly, the Kelvin-Helmholtz (KH) instability has been extensively studied at the magnetopause
and ionopause. However, there is no evidence on its influence on the bow shock oscillations. The
KH instabilities at the magnetopause can be easily excited and persistently exist as solar wind
streams with hundreds of kilometers per hour continuously pass through the magnetosphere.
Logically speaking, if this was a cause at the bow shock, the multiple-crossings should be a frequent
phenomenon that can be easily observed. But it is not the truth in observations.

Secondly, the plasma perturbations near the shock caused by the KH instability may be similar to
shock ripples, which can be distinguished by large changes of the shock normal directions. In the
revised version, we calculate the shock normal directions which could be found in Extended Data
Tables 1 and 2. One can see that, the shock normal directions change little in our two cases.

(2) Magnetic reconnections in the tail

Thanks for this idea. We cannot validate this speculation unless we have a third spacecraft right in
the tail. We include it in the possible mechanisms in the manuscript for future examination (Lines
223-230). The magnetic reconnection in the tail is also kind of the internal driver in the
magnetosphere.

(3) Solar wind background

The solar wind background is also an import factor for future studies, as we anticipate that the
internal processes that may cause BS oscillations may depend on solar wind conditions.

In the revised manuscript, we analyzed the IMF properties for each event. Specifically, we
calculated the angle between the IMF and the shock normal, confirming that both events involved
quasi-perpendicular shocks. The IMF has a positive component of B_x and negative B_y , suggesting a
negative polarity of IMF (Lines 127-131). We examined the IMF clock angles (determined by
$\text{acos}(B_y/B_z)$), which indicated a $-B_y$ orientation. A discussion about the possible effect of IMF clock
angle on the phenomenon is added in Lines 201-222.

We also found that both events occurred during the steady phase of slow solar wind streams, with
solar wind speeds of ~ 320 km/s and ~ 340 km/s, respectively (Lines 175-176).

(4) Extensive multiple bow shock crossing observations

Multiple bow shock crossing observations have been extensively report before. It is intuitive to
understand the influences of solar wind perturbations and shock transient structures on Mars' bow
shock. However, the internal drivers have not been reported. Hence, we believe that it is the first
report of the unexpected and new phenomenon that may stimulate follow-up studies for the
mechanisms behind the phenomenon.

Reviewer #1 (Remarks to the Author):

Thank you for addressing many of the concerns that were raised in the previous review. The inclusion of MVA along the boundary and a discussion of the upstream IMF with respect to the shock and regarding magnetic reconnection significantly improve the paper.

I completely agree with the authors that this is a compelling study that is worth further investigation. The two events presented here are indeed quite interesting and I agree with the authors that they show evidence for internal dynamics driving bow shock oscillations.

However, the explanations for the observed oscillations and the limited number of events (only two) make the conclusions of this manuscript too speculative for publication in its current form. In my previous review, I asked whether there is evidence of reconnection affecting bow shocks at other planets, such as Earth. Such evidence would strongly support the argument that similar processes might occur at Mars and would reduce the level of speculation required to substantiate the conclusions presented here. It appears that such evidence is lacking, which makes the current argument more speculative.

I recently came across Garnier et al., 2022 that discusses the influence of crustal fields on the Martian bow shock location. It is possible that the crustal field-reconnection argument could be strengthened with references to the conclusions presented in that manuscript. If the authors agree, I would encourage them to resubmit one more time with a more robust argument that reconnection could be driving bow shock oscillations. In its present form, the argument is too speculative to be published here.

Reply: Thank you for your encouraging comments. Yes. We agree that the argument is too speculative to be publishable. So driven by you and the other reviewer, we re-examine the data, conduct a further analysis and numerical simulations, and conclude that the observed BS oscillations were due to the weakness of the bow shock, characterized by a low Mach number, and the fluctuations of solar wind parameters. Please see the section “Cause of the Oscillations” for details.

Figure 3 and Lines 156-164: I am confused by panel g. We see a big change in magnetosonic mach number, which you explain is part of the SWIA mode switch. Magnetosonic mach numbers < 2 in the solar wind at Mars' orbit would be extremely rare, which makes me doubt the accuracy of this prediction. Can you explain more why we may see the jump in Mach number and how accurate these predictions are?

Reply: The SWIA solar wind mode measures the solar wind ion flux with a high angular resolution but a narrower field of view (FOV), while the sheath mode measures the ion flux with a broader FOV but a lower angular resolution. The Mach number derived from plasma moments in SWIA sheath mode should be underestimated. So in the manuscript, we separate the two modes events (circles and diamonds in Fig.4 of the manuscript) when comparing multi-crossing events with single-crossing events. As shown in Fig. 4, the Mach number of Event 1 in the solar wind mode (red circle) is notably lower than those of single-crossing events in the same solar wind mode (black circles), whereas the Mach number of Event 2 in the sheath mode (red diamond) is also lower than

those of single-crossing events in the same sheath mode (black diamonds).

Line 180: I recently came across Garnier et al., 2022 that investigated the influence of Mars crustal fields on bow shock location. This investigation has important implications for your work, I would strongly recommend that you cite this paper and discuss its implications here.

Reply: Thank you for the reminding. Garnier et al. (2022) is mentioned at Line 56.

Line 187-191: It seems to me that the presence of single BS crossings under steady SW conditions near longitudes of the multiple crossing events (i.e. black triangles near red circles in Figure 4) suggests that the BS is not always variable, even under similar subsolar longitude conditions. Is this the correct interpretation? To me, this suggests that it may be some interaction between the crustal fields and upstream conditions that leads to variable BS events. Am I understanding correctly?

Reply: Yes. Your understanding is correct. We found that the BS subsolar longitudes during single-crossing events are nearly uniformly distributed, which means there were also single-crossing events when the subsolar longitudes were around the west boundary of the Martian strongest magnetic anomaly region. So it actually suggests that the Martian crustal field is not likely the cause, or at least not the only cause, for the observed BS oscillations as we now state at Lines 179-240 in the manuscript.

Reviewer #2 (Remarks to the Author):

Second review of Internally Driven Oscillations of Martian Bow Shock by Cheng et al.

This paper focuses on bow shock variability in the flank of Mars during solar wind steady conditions and the potential for internal drivers to be the cause for this variability. The work is based on simultaneous observations of Tianwen-1 and MAVEN.

I appreciate very much the authors dedication with this work, which topic is very important for our understanding of Martian magnetospheric dynamics. However, my review is still the same as it was the first time. I do not think there is enough evidence in this paper to prove the authors' claim "it is the first report of the unexpected and new phenomenon that may stimulate follow-up studies for the mechanisms behind the phenomenon.". There is a long discussion in the paper about "speculative" arguments (this word even appears twice!) and none of them based on an analysis of the physics behind the oscillations. There is also a Table with plots from other papers that are out of context as we do not know the conditions where those observations were taken, or if they are comparable to the observations of this work.

Reply: Thank you! After the re-examination, we accept your critical comments. Our conclusion regarding internally driven oscillation was too speculative. In the revised manuscript, we have re-analyzed the upstream solar wind conditions and performed the 3D numerical simulations. It is found that the observed BS oscillations were due to the weakness of the bow shock, characterized by a low Mach number, and the fluctuations of solar wind parameters. Please see the section "Cause of the Oscillations" for details.

Table 1 presents the magnetic field and ion flux patterns of transients near the bow shock and is used to differentiate them from BS oscillations. We are not suggesting that these transients could not occur under the observed solar wind condition; rather, we aim to clarify that the observed multiple crossings are not the manifestation of these transients.

Moreover, there is not plot that shows the trajectories of both Tianwen and MAVEN. To me they look like at least MAVEN is skimming the bow shock, but it is difficult to get a robust conclusion without that basic information. This is the main reason that one can speculate with theories, but they do not demonstrate, in my opinion, that that they are the real reason behind the speculation.

Reply: The trajectories of both Tianwen-1 and MAVEN had been shown in Panels h and i in Figs. 1 and 3 in the manuscript, with the bow shock model included for reference. For Event 1, based on the discontinuities in the magnetic field and ion data from MAVEN observations (see Panels d–g of Fig. 1), MAVEN was skimming the bow shock at approximately 05:35. However, around t_3 and t_4 —when Tianwen-1 observed the oscillating bow shock—MAVEN was located in the upstream solar wind. For Event 2, MAVEN remained in the upstream solar wind throughout the period of interest.

I would like to insist that the topic is very interesting, and certainly I would be in favour of giving a more positive review if authors can prove their claims based on observations, theoretical analysis or numerical simulations that what they speculate is the real reason. Also, I would appreciate if replies would be included in the paper and not only to this referee, as other readers may have the same concerns.

Reply: Thank you. We have conducted a more detailed analysis of the upstream solar wind conditions during the events of interest as well as the numerical simulations. We change the conclusion to that the weak bow shock, under low Mach number conditions, may oscillate in response to weak solar wind disturbances. Please see the section “Cause of the Oscillations” for details.

We apologize for not including all our previous responses in the manuscript. In the revised version, we have added necessary discussions, e.g., the Kelvin-Helmholtz instability (Lines 134-139). Additionally, we have significantly expanded our analysis of the solar wind background.

Dear Editor and Reviewers,

Thank you for the insightful comments and suggestions. The largest change is that we add one more event, Event 37, for simulation to strengthen our conclusions (see updated Fig.5 and related text) as required by the first reviewer. Please find our detailed responses below, highlighted in blue. Corresponding changes in the manuscript are also marked in blue for your convenience. Hope the manuscript is now satisfactory.

Reviewer #1 (Remarks to the Author):

The latest version of manuscript is a significant change to the previous versions, focusing instead on the impact of magnetosonic Mach number of the susceptibility of the bow shock. They argue that lower upstream Mach numbers cause large-scale oscillations of the Martian bow shock, which could explain why Tianwen-1 observed multiple bow shock crossings while MAVEN measured relatively steady upstream conditions. The inclusion of BATS-R-US simulations of the bow shock are a welcome inclusion to the study, and show promise to support these conclusions.

However, I still find that the conclusions of this study are not adequately supported by the results as they are presented here. The multi-crossing events do seem to take place under low Mach numbers than the single crossing events, but the difference between these two values (4.7 vs. 5.0) is not very significant. It is still possible that this small change in Mach number could explain the results, but this is not demonstrated in the study as it is written currently. To show that this small difference in Mach numbers leads to a different number of crossings, one would need to run two simulations—one with a Mach number of 4.7 and another with 5.0—and demonstrate that this alters the bow shock oscillation behavior observed by Tianwen-1.

Reply 1: Thank you for the suggestion. We now add one more event for simulation—Event 37 with a Mach number of 5.0. As shown in Figure 5f and g., compared with the Event 1, the simulation of Event 37 does not show notable oscillations, leading to the BS single-crossing of Tianwen-1, consistent to the observations.

Furthermore, the simulations appear promising in illustrating the differences in bow shock oscillations between Tianwen-1 and MAVEN, but these differences are difficult to discern in their current presentation. The movies would be much clearer if they displayed the shock position flattened along the horizontal axis, highlighting how the boundary motion varies at different locations along the shock—similar to panels d and e in Figure 5.

Reply 2: Our intention was not to compare bow shock oscillations observed between Tianwen-1 and MAVEN. In this study, when Tianwen-1 was crossing the BS, MAVEN was in the upstream solar wind. MAVEN data are used to learn what causes the BS oscillations observed by Tianwen-1. From the supplementary movie, we can see that the multi-crossings of BS can be reproduced by the global BS oscillations.

Including a single crossing event (Event 12) as a comparison to the multi-crossing events is a good idea. However, additional details about the MHD simulation for Event 12 are needed to substantiate the study's conclusions. Specifically, the authors should include a time series—comparable to Figures 5b and 5c—showing both the observations and simulation results for Event 12, in order to clearly demonstrate that one scenario leads to multiple crossings at Tianwen-1 while the other results in only a single crossing.

Reply 3: We add the time series profiles of Event 12 in Figure 5d and e. The simulation roughly captures the BS crossing of Tianwen-1 though the amplitude of the magnetic field after the crossing is not as high as observations. The simulated BS position shown in Fig.5e shows one small oscillation of $\sim 0.1 R_M$ around 04:42, much smaller than multiple oscillations of about $0.2 R_M$ shown in Fig.5c for Event 1. After then, the BS of Event 12 continuously shrinks without notable oscillations, leading to a single crossing event recorded by Tianwen-1.

While I still agree that these events show promise for illuminating some interesting bow shock physics, the conclusions presented here are still not adequately supported by the results.

Reply 4: Thank you for your comment. To strengthen our conclusions, we conducted simulations for Event 1 (a multi-crossing event with a Mach number of 4.7), Event 12 (a single-crossing event under higher disturbance level of solar wind) and Event 37 (a single-crossing event with a slightly higher Mach number of 5.0) by following your suggestions. Hope these additions make the study more solid.

Lines 111: The bow shock also heats plasma in the magnetosheath compared to the solar wind. This would also broaden the distribution of protons and increase counts in the limited FOV.

Reply 5: We have mentioned the heat of plasma additionally in Lines 109-111.

Line 193: Please clarify what “angular angle” means.

Reply 6: Sorry for the confusion. We have replaced “angular angle” with “angular extent,” referring to the angle between the bow shock location and the crustal magnetic fields with respect to the Mars center. This clarification has been made in the revised manuscript in Lines 192-193.

Lines 229-230: The difference between a Mach number of 4.7 vs. a Mach number of 5.0 is not very significant. In order to demonstrate that this small difference in Mach numbers result in a different number of crossings, one would need to produce two simulations (one with Mach number of 4.7 and another with Mach number of 5.0) and demonstrate that this would change bow shock oscillation behaviour at Tianwen-1.

Reply 7: Added. See Fig.5f and g, and associated text. Also see Replies 1 and 4.

Line 245-246: This choice to shift the simulation results by 15 and 30 minutes should be justified and discussed in greater detail. The similarities between Case 1 and Case 2 (Figure 5b and 5c) is somewhat misleading because the different shifts of the model results suggest the bow shock does behave differently given different crustal field locations.

Reply 8: We now discuss the time shifts in the Method in the revised manuscript (Lines 644-652).

In Case 1, the strongest Martian crustal magnetic fields are located on the duskside, while in Case 2, Mars was rotated to position the strongest crustal fields on the nightside. The difference in magnetic pressure between these two configurations leads to a shift in the overall bow shock position, which accounts for the applied 15- and 30-minute time adjustments when aligning the simulation results with the observations.

Despite the difference in crustal field distribution, both simulations show a highly oscillatory bow shock, suggesting that the oscillations are primarily driven by upstream solar wind conditions rather than the specific location of the crustal field or the rotation of the crustal field. Previous statistical studies (e.g., Garnier et al., 2022) have revealed that the crustal field can alter the position of BS. But the time scale of such alterations is hours rather than the minutes investigated in this study.

Figure 4b: It is difficult to distinguish the diamonds from the circles. I would suggest making this difference between SWIA modes clearer. Maybe circles and triangles?

Reply 9: Thanks. We have modified the figure by filling the diamond symbols with gray.

Lines 256-259: I believe that the choice for Event 12 as a useful comparison requires more justification. Is Event 12 also a quasi-perpendicular shock? Does this conclusion hold for other events with similar Mach numbers?

Reply 10: Thanks for the comment.

- (1) Event 12 is selected because Tianwen-1 crossed the bow shock at a location similar to Event 1 and the solar wind disturbance level is larger than that of Event 1. So that we can see if larger disturbance level can cause BS oscillations. The results reveal that the low disturbance level under the low Mach number can more easily result in the bow shock oscillation than the large disturbance level with the high Mach number, suggesting the importance of Mach number.
- (2) Event 12 is indeed a quasi-perpendicular shock, with a θ_{Bn} of approximately 80° .
- (3) Not all events with similar Mach numbers exhibit a quasi-perpendicular configuration. For example, Event 37 has a similarly low Mach number (5.0) and corresponds to a quasi-parallel shock, with a θ_{Bn} of approximately 30° .

Line 260: The difference between Movies 1 and 2 is very difficult to see at this scale. The format of these movies should be changed to better demonstrate the point the authors are trying to make.

Reply 11: Now we only leave one movie to demonstrate the overall picture of the simulations. The differences among simulated events have been clearly revealed in Figure 5b-g.

Figure 5: It is necessary to compare this with the simulated data from the single crossing event (Event 12) to demonstrate that larger Mach numbers suppress these oscillations. It is important to include another time series (similar to 5b and 5c) but for the observations and simulated results of Event 12.

Reply 12: Thanks. Added as Fig.5d.

Supplementary Movies 1 and 2: The simulations show promise for demonstrating the difference in oscillation of the bow shock at Tianwen-1 vs. MAVEN, but this is quite difficult to see in its current form. These movies would greatly benefit from showing the shock position flattened out on the horizontal axis and showing how this boundary motion is different at different points along the shock (similar to panels d and e in Figure 5).

Reply 13: Please see Reply 2. We never compare the oscillation between Tianwen-1 and MAVEN. MAVEN is used to monitor the upstream solar wind conditions.

Lines 266-268: Mach number doesn't only depend on the speed of the solar wind. Based on your argument, would large IMF $|B|$ also lead to bow shock oscillations because Mach number is inversely proportional to $|B|$? What about solar wind density?

Reply 14: Thank you for the comment. The values of solar wind density and $|B|$ are shown in Extended Data Figure 3d and 3e, respectively. Based on our analysis, we do not observe a clear correlation between bow shock oscillations and either solar wind density or $|B|$. That means Mach number as a combined parameter is a better indicator than the elemental parameters, like, the density or $|B|$.

Reviewer #2 (Remarks to the Author):

Review of Oscillations of Mars' Bow Shock Under Weakly Disturbed Solar Wind Conditions by Cheng et al.

This paper focuses on bow shock variability in the flank of Mars during steady conditions and the potential for internal drivers to be the cause for this variability. The work is based on simultaneous observations of Tianwen-1 and MAVEN and simulations.

The work has significantly improved since last version. The new simulation adds a great value and make the conclusions much more evident than in the previous versions. The discussion is also good and appropriate.

Thanks for your positive feedback.

I only have a few minor suggestions in this round:

- In the abstract and in other parts of the text: I don't think it is appropriate to say "the Mach number is probably the most sensitive..." the word "probably" reduces credibility to the results.

Reply 15: We remove the word "probably" in the manuscript.

- The 35 crossings that are compared to the two selected, how they were chosen? From Table 4 I understand they come from similar periods of time, but this is not very obvious to infer from the paper. Can authors add more clarifications on this aspect in Section "cause of the oscillations"? Are all those cases comparable in terms of solar wind activity, IMF direction, Mach number, Tianwen-1 and MAVEN locations, latitude, etc?

Reply 16: These single-crossing events are selected from the same period based on the following criteria of (1) the clear signature of the BS crossing in magnetic field from Tianwen-1, (2) no notable fluctuations blurring the distinction between single- and multiple-crossings, and (3) availability of the simultaneous observations of the upstream solar wind from MAVEN. We clarify the selection at Lines 199-204. In the one and a half months, Tianwen-1 and MAVEN's orbits did not change too much. The solar wind parameters, including IMF direction, Mach number, etc., of these events are analyzed in Fig.4b and Extended Data Fig.3.

- Figure 4 is a bit confusing. Which ones are the crossings of Figures 2 and 3? Not obvious. Also, I see at least 8 crossings with similar low Mach number and many others with similar "higher" Mach number. How the authors conclude that this parameter is a key to characterise large-scale disturbances? The plot is not obvious to interpret, and therefore, the conclusions are difficult to

take. Have all these 35 events been analysed in order to characterise the flapping of the magnetosphere?

Reply 17: Sorry for the confusion. The red circle represents Event 1 shown by Figure 1, while the red diamond represents Event 2 shown by Figure 2. We have clarified them in the caption of Figure 4. The 8 low Mach number events (diamonds) in Fig.4b were observed in different mode by SWIA from the many other higher Mach number events (circles). The former is in sheath mode and the latter in solar wind mode. This was clarified at Lines 230-233, 236-239, and in the Caption of Fig.4 at Lines 302-303.

All the 35 events are analyzed observationally as shown in Fig.4 and Extended Data Fig.3. But not all of them are simulated due to the huge consumption of computing resources. In this version, we add one more event for simulation: Event 37 which has a Mach number 5.0, slightly higher than Event 1 with the Mach number 4.7. We think that the two control cases (Event 37 and Event 12) further strengthen the study.

- I still don't see the trajectories of both MAVEN and Tianwen-1 in Figures 1 and 3, and the interpretation of the results is difficult. I only see the transit of the data that have been plotted. Ideally, it would be good to have the whole orbit plotted.

Reply 18: Thank you for the suggestion. We have updated Figures 1h-i and 3h-i to include the full trajectories of both MAVEN and Tianwen-1.

- Are the numerical simulations appropriate? The pink profiles in Figure 5 show some discrepancies with Tianwen-1 blue profiles, and not sure it really captures the "basic observational features" as where the data has a dip, the simulation has an increase, an viceversa. Are these discrepancies expected and acknowledged? It deserves more explanations in the paper.

Reply 19: Thank you for the comment. We have pointed out this issue in the main text at Lines 250-253, and made a discussion in Methods (Lines 644-659) as below:

"Due to the limitation of the computational resource on the spatial and temporal resolutions, the BS location in the simulation differs from the observation and the magnetic field fluctuations are not as large as observations."

"Given that the simulations are MHD-based and thus limited in capturing small-scale or transient features, such as fine BS structures, wave activity, or short-timescale fluctuations, the differences between simulations and observations are expected. Despite these discrepancies, the simulations reproduce the key large-scale characteristics relevant to our study, i.e., the overall amplitude and presence of BS oscillations under weakly perturbed solar wind conditions. The general trend and level of oscillatory behavior are consistent with the observations and support our interpretation."

- Table 1: same comment as in the previous revision. What's the point of this table when the conditions are not comparable? At least I would remove the plots within the table as they only

introduce confusion and keep the last 3 rows, and please explain all the acronyms in that figure, like for example, what is SLAMS? Or HFA/SHFA? Although it is said in the text, I believe the table should be self-explanatory.

Reply 20: We have removed the plots and acronyms in the table.